# DataSciBench: An LLM Agent Benchmark for Data Science

## Abstract

This paper presents `DataSciBench`, a comprehensive benchmark for evaluating Large Language Model (LLM) capabilities in data science. Recent related benchmarks have primarily focused on single tasks, easily obtainable ground truth, and straightforward evaluation metrics, which limits the scope of tasks that can be evaluated. In contrast, `DataSciBench` is constructed based on a more comprehensive and curated collection of natural and challenging prompts. We develop a semi-automated pipeline for generating ground truth (GT) and validating evaluation metrics. This pipeline utilizes and implements an LLM-based self-consistency strategy to produce accurate GT by leveraging collected prompts, predefined task types, and aggregate metrics. Furthermore, it employs a careful approach to filter a high-quality Task - Function - Code (TFC) list and assess each code execution outcome within TFC based on precisely defined metrics and programmatic rules. Our experimental framework involves testing 6 API-based models, 8 open-source general models, and 9 open-source code generation models using the diverse set of prompts we have gathered. Through this approach, we aim to provide a more comprehensive and rigorous evaluation of LLMs in the domain of data science, shedding light on their strengths and weaknesses. Experimental results demonstrate that API-based models greatly outperform open-sourced models on all metrics except for VLM-as-a-judge and Deepseek-Coder-33B-Instruct achieves the highest score among open-sourced models. We release all code and data at code.

## 1 Introduction

Large language models (LLMs) (Achiam et al., 2023; Team et al., 2023; GLM et al., 2024) are increasingly used in data science and scientific domains, e.g., data analysis (Hong et al., 2024), protein generation (Jumper et al., 2021; Chen et al., 2024), and scientific reasoning (Zhang et al., 2024a;b). For data science tasks, LLMs offer the potential to (semi-)autonomously conduct data analysis (Huang et al., 2023) and data visualization (Hong et al., 2024) by calling code interpreters with corresponding Python libraries given the public known problems. These works are benchmarked on relatively straightforward tasks where ground truth labels can be precisely obtained. However, much of real-world data analysis requires reasoning over more complex scenarios, such as evaluating the quality of the images generated by the data visualization task. The proper evaluation of these more complex data science tasks remains an open research direction.

While there are some existing benchmarks used to evaluate LLMs for related challenges (see Table 1), those benchmarks typically focus on evaluating narrower tasks with easy-to-obtain ground truth and straightforward evaluation metrics. For example, MLAgentBench (Huang et al., 2023) presents a machine learning research benchmark by building an LLM Agent pipeline. SWE-Bench (Jimenez et al., 2023) benchmarks the abilities of LLM to solve real-world software issues from GitHub. InfiAgent-DABench (Hu et al., 2024) completes data analysis tasks by generating labels with GPT-4 and calculating accuracy. The frontier of LLM evaluation is towards more complex real-world tasks that consist of multiple subtasks. For these challenging prompts, how to generate ground truth and define specific evaluation metrics for each subtask in a comprehensive perspective is a question worth exploring.

Table 1: Comparison with related work. LC denotes LeetCode.

| Benchmark | Prompt Source | Evaluation Metrics |
|---|---|---|
| DS-1000 (Lai et al., 2023) | StackOverflow | Test Cases + Surface-Form Constraints |
| MLAgentBench (Huang et al., 2023) | Kaggle | Acc. + Success Rate, Human |
| LiveCodeBench (Jain et al., 2024) | LC & AtCoder & CodeForces | Test Cases + Pass Rate |
| NaturalCodeBench (Zhang et al., 2024c) | CodeGeeX | Test Cases + Pass Rate |
| BigCodeBench (Zhuo et al., 2024) | StackOverflow | Test Cases + Pass Rate |
| Text2Analysis (He et al., 2023) | Human & LLM | Executable code ratio, Acc., Regression scores |
| InfiAgent-DABench (Hu et al., 2024) | LLM | Acc. |
| DataSciBench (Ours) | Human & CodeGeeX & BCB | Aggregate Metrics and Programmatic Rules |

In this paper, we introduce a new benchmark, called DataSciBench, which evaluates the data science abilities of LLMs and helps LLMs improve their data analysis and data visualization abilities. Regarding collected prompts, their corresponding responses, and evaluation metrics, we hope that they meet the following characteristics: (1) Require more natural, challenging, and high-quality prompts to promote the development of LLMs' improvement. (2) Strong correlations are necessary for sequential tasks so that models can be distinguished well. (3) Multiple types of results are required to perform comprehensive evaluations.

To bridge the gap between task definition, evaluation criteria, and automated assessment in data science contexts, we propose a novel semi-automated framework, called Task - Function - Code (TFC) generation and evaluation. Specifically, from coarse-grained perspectives, we first aggregate the range of task types, functions, and corresponding codes, then, from fine-grained perspectives, we define programmatic rules for the outputs of each function depending on the specific tasks and compare results with ground truth to ensure fair and consistent assessment. To validate the effectiveness of LLMs and our proposed TFC pipeline on our collected comprehensive prompts, we experiment with 6 API-based models, 8 open-sourced general models, and 9 open-sourced code generation models. We observe that API-based models greatly outperform open-sourced models on average. Specifically, GPT-4o surpasses all other models on all metrics except for VLM-as-a-judge and Deepseek-Coder-33B-Instruct achieves the highest score among open-sourced models. However, all models have significant room for improvement in following fine-grained instructions, calling the appropriate tools, executing accurate plans, and exporting the required execution outputs.

Overall, our key contributions are as follows:

- We introduce DataSciBench, a comprehensive benchmark designed to assess the performance of LLMs in data science tasks. We develop a semi-automated pipeline to generate ground truth and evaluate aggregated metrics using carefully crafted complex questions.

- We propose a Task-Function-Code (TFC) list based on predefined aggregated metrics and programmatic rules. We then assess 23 large language models from both coarse-grained and fine-grained perspectives, presenting the results in Table 2.

- Our study includes various analyses such as comparisons and correlations with existing benchmarks, presented in Figure 5 and Table 3. Furthermore, we offer research insights derived from experimental outcomes of the evaluated LLMs that point to interesting directions for future work.

## 2 BACKGROUND ON USING LLMs FOR DATA SCIENCE

This section discusses the key aspects that underlie our benchmarking approach.

**Ground Truth Generation.** Ground truth serves as the cornerstone for evaluating the performance of LLMs in data science tasks. For diverse and challenging data science prompts, we aim to propose a semi-automated pipeline that leverages a robust LLM to generate ground truth and employs a self-consistency strategy to ensure the accuracy and reliability of the generated ground truth.

**Evaluation Metrics Definition.** Defining appropriate and meaningful evaluation metrics is essential for effectively comparing and analyzing the effectiveness of different LLMs in data science tasks. In our study, we meticulously define evaluation metrics tailored to the specific tasks and challenges

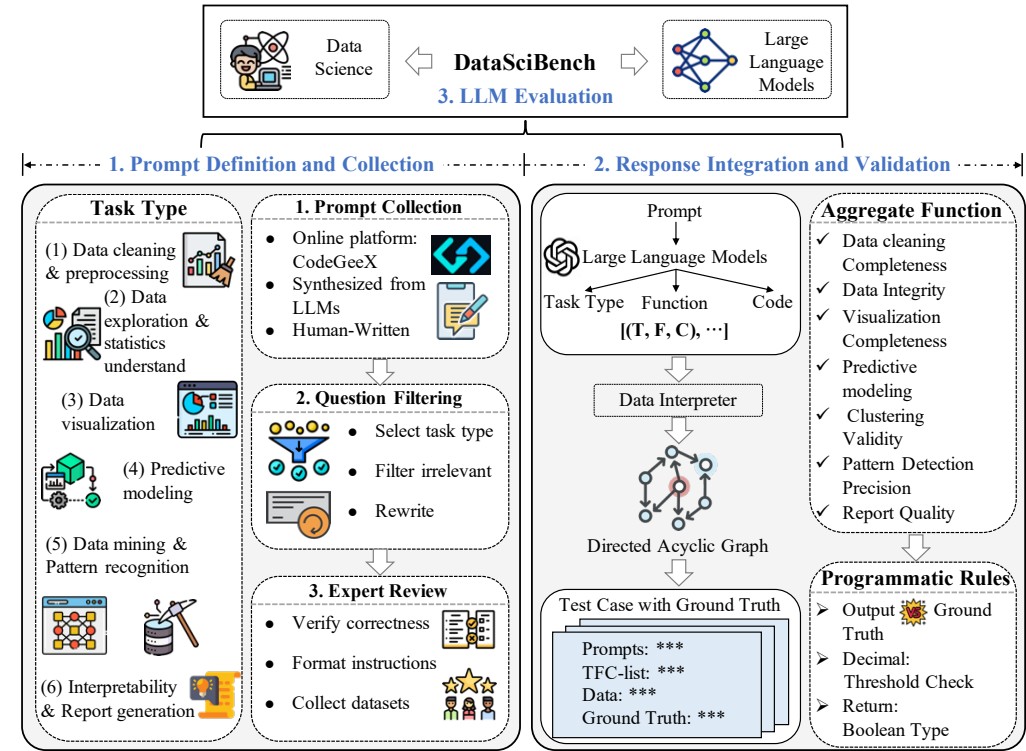

Figure 1: The overall framework of `DataSciBench` includes prompt definition and collection, response integration and validation, and LLM evaluation.

posed by the collected prompts. These metrics are designed to capture the diverse nuances of data analysis and visualization tasks, enabling a comprehensive assessment of LLMs' capabilities.

**Limitation of Previous Studies.** Prior research in benchmarking LLMs for data science has often been limited by focusing on single tasks, simplistic evaluation metrics, and readily available ground truth. These shortcomings hinder the thorough evaluation of LLMs and may not fully capture their strengths and weaknesses. By addressing these limitations, our study seeks to provide a more comprehensive and nuanced evaluation of LLMs in data science applications. Through the development of `DataSciBench` and the implementation of a rigorous evaluation framework, we aim to push the boundaries of benchmarking practices in the field of data science and LLM research.

## 3 DATASCIBENCH

`DataSciBench` consists of three important components as outlined in Figure 1.

- **Prompt Definition and Collection** which defines 6 task types and collects 222 real, challenging, and high-quality prompts through question filtering and expert review.

- **Response Integration and Validation** which proposes novel Task - Function - Code (TFC) to assess the key tasks for each prompt and defines the aggregate functions and programmatic rules to effectively evaluate the specific task description and compare prediction with ground truth.

- **LLM Evaluation** which assesses 6 API-based models, 8 open-sourced general models, and 9 open-sourced code generation models from coarse-grained (e.g., success rate, completion rate) and fine-grained (e.g., VLM-as-a-judge, aggregate functions) perspectives.

### 3.1 PROMPT DEFINITION FOR DATA SCIENCE

**Task Type.** We define six typical data science tasks as follows:

1. **Data cleaning and preprocessing.** This task detects and processes missing values, outliers, and duplicate data; and standardizes data formats, such as a uniform format for dates and times.

2. **Data exploration and statistics understand.** This task calculates basic statistical indicators of data (mean, median, standard deviation, etc.), generates data distribution charts (histograms, box plots, etc.), calculates correlations between variables, and draws correlation matrices or maps.

3. **Data visualization.** The goal of this task is to visualize and analyze data and create interactive charts so users can freely explore the data.

4. **Predictive modeling.** The task aims to select the appropriate machine learning algorithm, such as linear regression, decision tree, random forest, etc.; carry out feature engineering, such as feature selection, feature transformation, feature combination, etc.; the data set is divided into the training set and test set, and the model is trained and evaluated; and select the corresponding evaluation indicators for different prediction problems, such as classification, regression or clustering.

5. **Data mining and Pattern recognition.** This task uses association rule mining, frequent item set mining, and other methods to find interesting patterns in the data; Text mining technology is used to extract keywords, topics, and other information from text data; and apply cluster analysis, classification algorithms, etc. to identify underlying patterns and structures. Pattern recognition tasks can conduct these functions: image recognition, text clustering, and time series detection.

6. **Interpretability and Report generation**. This task aims to provide explanations of model results, such as feature importance, model parameters, etc., and automatically generate reports and summaries that present the results of the analysis in a way that is easy to understand and share.

**Task Integration.** To increase the difficulty of assessing the prompt, we chose more complex prompts that included multiple tasks. These sequential tasks can be any combination of tasks.

## 3.2 DATASET COLLECTION

**Question Collection.** We collect questions from four sources:

- Coarse-grained collection from a real-world online platform. We collect natural prompts from one online code-generation platform, CodeGeeX (Zheng et al., 2023).

- Extracted and rewritten from a public code benchmark. We select data science-related and high-quality prompts from BigCodeBench and then rewrite them to unified instructions.

- Hand-Written by humans. We also write elaborated prompts to increase the difficulty and robustness of the evaluated benchmark by referring to relative websites[1].

- Synthesized from LLMs. We use a few-shot examples drawn from human-written prompts to ask LLM to generate similar prompts.

**Question Filtering.** We filter low-quality questions via the following three principles: (1) Choose questions that keywords include, but are not limited to, "machine learning", "deep learning", "data preprocessing", and "data visualization". (2) Filter questions that require updating code, finding errors, and explaining concepts. (3) Rewrite questions that align with human preferences and LLMs, which refers to questions solvable by both humans and large language models, avoiding overly specialized or ambiguous queries.

**Expert Review.** To ensure the quality of the collected prompts, we review them by experts who are professionals in computer science and data analysis. The review process includes three stages: **(1) In stage 1,** experts verify the correctness and adjust the suitability of prompts. In addition, experts ensure the answers to the prompts are easy to evaluate. For example, handing missing values for a data frame. **(2) In stage 2**, experts format all prompts into unified instructions and the format encompasses input data, input file, prompt, and expected output file. **(3) In stage 3**, experts ensure the availability of datasets of input prompts, including generating random datasets and collecting the public datasets.

---

[1]https://ds100.org/course-notes/eda/eda.html

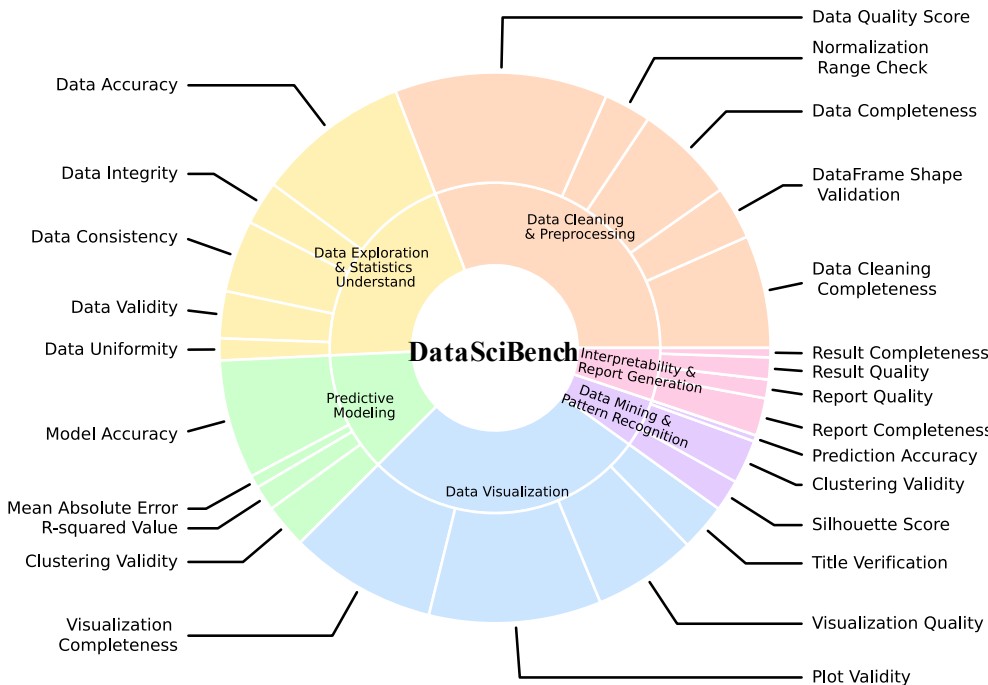

Figure 2: Statistics of task types and aggregate functions.

## 3.3 RESPONSE INTEGRATION AND VALIDATION

**Ground Truth Generation and Validation.** To obtain the response to collected prompts, we propose the following strategy to generate a test case of each prompt. We first obtain the outputs of each prompt by sampling LLMs several times and the final output by executing the generated code. We use two different validation methods to ensure the rationality and reliability of the answer generated by LLMs. With regard to prompts originating from BigCodeBench where reliable test cases are provided, we validate the generated answer by performing all test cases. Answers that pass all test cases are rechecked by humans and finally considered as ground truth. As for other prompts, we initially adopt a self-consistency strategy (Wang et al., 2022) to obtain generated codes, and then manually validate these results manually through cross-verification by multiple authors to ensure accuracy and reliability.

**Evaluation Selection.** We introduce a structured approach to identify and evaluate key tasks across six established types. We first use GPT-4o-mini to select several valuable task types, return corresponding evaluation functions, and generate the evaluation codes for each prompt to effectively evaluate the capabilities of LLMs and reduce the evaluation cost. Each group data is simplified as a tuple (T, F, C) in generated **R** as follows:

$$\mathbf{R} = \{(\mathbf{T}_i, \mathbf{F}_i, \mathbf{C}_i)|_i^N\}, \tag{1}$$

where $N$ is the number of valuable task types per prompt, and this value is different for each question. Then we conduct a data interpreter (DI) (Hong et al., 2024) to generate a directed acyclic graph (DAG) in a hierarchical structure for each prompt, in which each task type is defined as a node at one level in a DAG. Based on the generated graphs, we take a powerful LLM as a backbone and run all evaluation functions to obtain the ground truth of each task type. To some extent, this way of verification can avoid the commonly used LLM-as-a-Judge black-box assessment.

**Function Aggregation.** To unify the key functions and improve the scalability of the evaluation, we select top-K functions for each task type and aggregate all generated functions to the top-K function category, as shown in Figure 2. Generally, the K is set as 5. For instance, the function category for data cleaning and preprocessing includes Data Cleaning Completeness, DataFrame Shape Validation, Data Completeness, Normalization Range Check, and Data Quality Score.

**Programmatic Rules.** Regarding aggregate functions with corresponding codes, we define unified rules to validate generated code. Specifically, we unify all outputs as boolean or decimal types ranging between 0 and 1. Then, we obtain the final value by comparing ground truth with prediction output depending on the specific task description of aggregate functions. For example, regarding Data Cleaning Completeness, which calculates the final number of rows/columns after preprocessing, the final output is 1 if the number is the same as the number of ground truths otherwise 0. For some specific tasks whose output type is decimal, we also set a corresponding threshold to transform the output to boolean for simplicity, such as, the threshold being set to 0.5 if the aggregate function is silhouette score for data mining and pattern recognition.

**Summary.** Based on the abovementioned processes, we obtain 222 effective prompts and corresponding test cases, which help the following evaluations of API-based and open-sourced models.

## 4 EXPERIMENTS

### 4.1 SETTINGS

To assess the performance of different models (e.g., API-based models and open-sourced general/code generation models), we construct a comprehensive benchmark on our collected prompts.

- **Six API-based models** include o1-mini/GPT-4o-mini/GPT-4o-2024-05-13/GPT-4-Turbo (Achiam et al., 2023), Claude-3.5-Sonnet[2], and GLM-4-Flash (GLM et al., 2024).
- **Eight open-sourced general models** include Llama3.1-8B-Instruct, Llama3-8B-Instruct, Qwen2.5-7B-Instruct, Qwen2-1.5/7B-Instruct (Yang et al., 2024), Gemma2-9B-it (Team et al., 2024), GLM-4-9B-chat (GLM et al., 2024), Yi-1.5-9B-chat-16k (Young et al., 2024).
- **Nine open-sourced code generation models** include Deepseek-Coder-1.3/6.7/33B-Instruct (Guo et al., 2024), CodeLlama-7/13/34B-Instruct (Roziere et al., 2023), Qwen2.5-Coder-1.5/7B-Instruct (Hui et al., 2024), and StarCoder2-15B (Lozhkov et al., 2024).

### 4.2 EVALUATION METRICS

**Coarse-grained Metrics.** We define the coarse-grained metrics (CR and SR) for evaluating LLMs.

• **Completion Rate (CR).** Following Data Interpreter (Hong et al., 2024), we calculate the Completion Rate given our TFC. For each TFC in the TFC list, we give it a completion score, with a minimum score of 0 and a maximum score of 2. The step completion scores were given as follows: missing (score of 0), fail (score of 0), success-non-compliant (score of 1), and success-compliant (score of 2). The final completion rate is then calculated as follows:

$$\text{Completion Rate (CR)} = \frac{\sum_{t=1}^{T} s_t}{T \times s_{\max}}, \quad (2)$$

where the numerator was the sum of the completion scores for each step, and the denominator was the sum of the maximum possible scores for all steps ($2 \times T$ and $T$ is the number of TFCs).

• **Success Rate (SR).** Similar to Codex (Chen et al., 2021), our success rate is defined as the rate of complete success on a single prompt estimated under 10 runs. Specifically, if all the TFCs have passed within a run of a single prompt, it will count as a success. Otherwise, it will count as a failure. Note that for prompts acquired from BigCodeBench, we compare the completion function's outputs with the ground truth completion function's outputs to determine whether a single run passes, since TFCs are derived based on demanded function outputs in this case. The formula for calculating SR is as follows:

$$\text{Success Rate (SR)} := \mathop{\mathbb{E}}_{\text{Prompts}} \left[ 1 - \frac{\binom{n-c}{k}}{\binom{n}{k}} \right], \quad (3)$$

where $n = 10$ and $k = 1$ in our case, $c$ refers to the number of runs that have passed all the TFCs.

---

[2]https://www.anthropic.com/news/claude-3-5-sonnet

Table 2: Overall evaluation results for `DataSciBench` on all our curated prompts.

| Models | Size | Coarse-grained Metrics | | Fine-grained Metrics | | | | | | Score |
|---|---|---|---|---|---|---|---|---|---|---|
| | | SR (%) | CR (%) | VLM | F1 | F2 | F3 | F4 | F5 | |
| o1-mini | N/A | 29.77 | 45.26 | 1.75 | 44.63 | 19.27 | 36.01 | 30.94 | 23.81 | 38.73 |
| GPT-4o-2024-05-13 | N/A | **66.31** | **68.44** | 2.10 | **75.93** | **56.14** | **69.33** | **71.35** | **57.67** | **64.43** |
| GPT-4o-mini | N/A | 50.63 | 57.78 | 1.65 | 60.30 | 48.02 | 57.84 | 59.24 | 53.54 | 54.12 |
| GPT-4-Turbo | N/A | 51.93 | 58.87 | 1.85 | 62.30 | 41.62 | 57.75 | 60.25 | 50.75 | 54.59 |
| Claude-3-5-Sonnet-20240620 | N/A | 47.48 | 58.11 | 1.44 | 49.07 | 36.94 | 55.84 | 52.87 | 46.04 | 52.26 |
| GLM-4-Flash | N/A | 30.32 | 34.04 | 1.51 | 36.53 | 29.42 | 32.57 | 27.64 | 14.44 | 30.75 |
| Meta-Llama-3.1-8B-Instruct | 8B | 24.73 | 33.89 | 1.55 | 38.24 | 18.25 | 21.98 | 22.89 | 25.85 | 29.70 |
| Meta-Llama-3-8B-Instruct | 8B | 2.88 | 3.92 | 1.93 | 4.18 | 1.26 | 2.70 | 2.67 | 1.47 | 3.40 |
| Gemma-2-9B-it | 9B | 7.07 | 11.00 | 1.63 | 26.16 | 16.90 | 23.81 | 18.11 | 17.15 | 12.69 |
| GLM-4-9B-Chat | 9B | 25.72 | 30.38 | 1.56 | 31.51 | 23.15 | 28.07 | 27.19 | 19.14 | 27.57 |
| Qwen2.5-7B-Instruct | 7B | 43.83 | 50.74 | 1.44 | 51.18 | 36.41 | 47.25 | 45.24 | 34.77 | 45.99 |
| Qwen2-7B-Instruct | 7B | 22.84 | 25.58 | 1.68 | 30.93 | 20.78 | 28.73 | 25.87 | 7.52 | 23.54 |
| Qwen2-1.5B-Instruct | 1.5B | 3.96 | 5.46 | 1.54 | 4.54 | 1.98 | 3.26 | 5.76 | 4.71 | 4.84 |
| Yi-1.5-9B-Chat-16K | 9B | 38.20 | 42.35 | 1.82 | 38.14 | 36.36 | 35.64 | 37.08 | 27.79 | 38.28 |
| CodeLlama-34B-Instruct | 34B | 0.90 | 1.47 | 0.00 | 1.02 | 0.84 | 1.98 | 1.54 | 1.19 | 1.33 |
| CodeLlama-13B-Instruct | 13B | 10.49 | 14.64 | **4.00** | 11.67 | 11.34 | 9.43 | 14.43 | 5.15 | 12.84 |
| CodeLlama-7B-Instruct | 7B | 2.88 | 3.97 | 0.00 | 3.53 | 2.37 | 2.57 | 1.74 | 1.59 | 3.32 |
| StarCoder2-15B | 15B | 2.07 | 2.61 | 2.33 | 2.57 | 1.81 | 1.59 | 3.43 | 1.19 | 2.45 |
| Deepseek-Coder-33B-instruct | 33B | 55.86 | 61.23 | 1.73 | 65.66 | 47.11 | 58.17 | 61.65 | 48.60 | 56.74 |
| Deepseek-Coder-6.7B-instruct | 6.7B | 37.03 | 41.62 | 1.30 | 43.49 | 34.57 | 46.36 | 46.49 | 18.09 | 38.42 |
| Deepseek-Coder-1.3B-instruct | 1.3B | 15.50 | 19.00 | 3.33 | 13.04 | 14.62 | 13.26 | 16.32 | 7.92 | 16.55 |
| Qwen2.5-Coder-7B-Instruct | 7B | 45.18 | 53.11 | 1.35 | 51.58 | 43.21 | 43.87 | 42.50 | 35.23 | 47.67 |
| Qwen2.5-Coder-1.5B-Instruct | 1.5B | 22.74 | 28.64 | 1.11 | 29.82 | 21.79 | 23.96 | 29.58 | 16.39 | 25.89 |

**Fine-grained Aggregate Metrics.** We also define the fine-grained aggregate metrics for detail evaluating all LLMs.

- Vision-language model (VLM)-as-a-judge assesses the overall score of two inputs based on pre-defined criteria (Appendix A.5), providing a step-by-step rationale for its evaluation.

- Data Quality Score (F1) in Data cleaning and preprocessing aims to assess the cleanliness of data post-preprocessing. It yields a boolean output of 1 if it matches the ground truth, or 0 otherwise.

- Plot Validity (F2) in Data visualization pertains to the accuracy of visual representations, such as checking whether the shape of the association matrix is consistent with the ground truth. If consistent, then the final value is 1, otherwise 0.

- Data Accuracy (F3) in Data Exploration and Statistics Understand focuses on understanding data quality and can be quantified using Mean Squared Error (MSE). The final value is derived by comparing it against the ground truth with a predefined threshold.

- Visualization Completeness (F4) in Data visualization evaluates the comprehensiveness of generated images (e.g., PNG, jpeg, PDF) by checking their existence compared to the ground truth. A score of 1 is assigned if the files exist, and 0 otherwise.

- Model Accuracy (F5) in Predictive modeling is utilized to gauge the predictive performance of models, providing a boolean accuracy value or decimal ranging between 0 and 1.

# 5 RESULTS AND ANALYSIS

## 5.1 OVERALL PERFORMANCE

We demonstrate overall experiment results in Table 2 and Figure 3. **(1) Concerning average performance, API-based models greatly outperform open-sourced models.** Among API-based models, GPT-4o achieves the highest total score of $64.43\%$, attaining a significant $9.84\%$ advantage over GPT-4-Turbo, which achieves $54.59\%$ total score. Remarkably, GPT-4o also surpasses all other models on all metrics except VLM, indicating its comprehensive capacity over various aspects. **(2) As for open-sourced models, the performance gap between general models and code generation models is insignificant.** Among those, Deepseek-Coder-33B-Instruct achieves the highest score of $56.74\%$, even outperforming various close-sourced models like o1-mini and GPT-4-Turbo. Other models like Qwen2.5-Coder-7B-Instruct and Qwen2.5-7B-Instruct also show fair good capability,

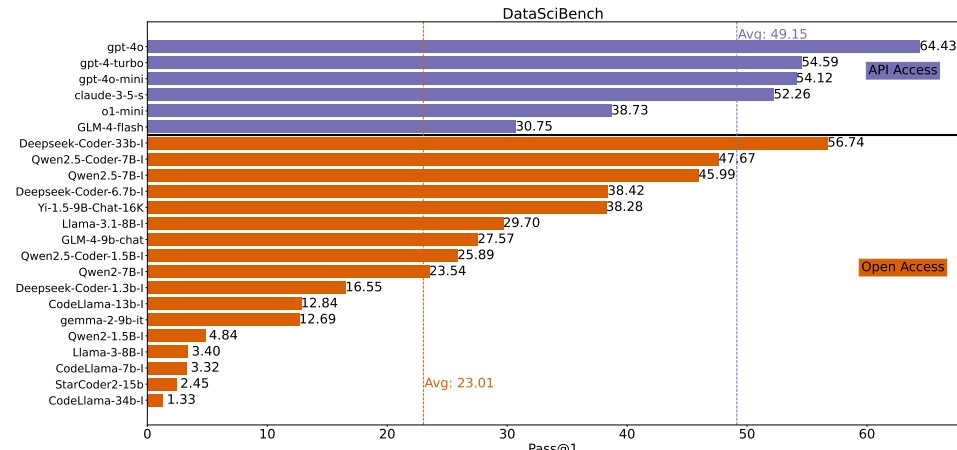

Figure 3: Overall score results of all tested LLMs.

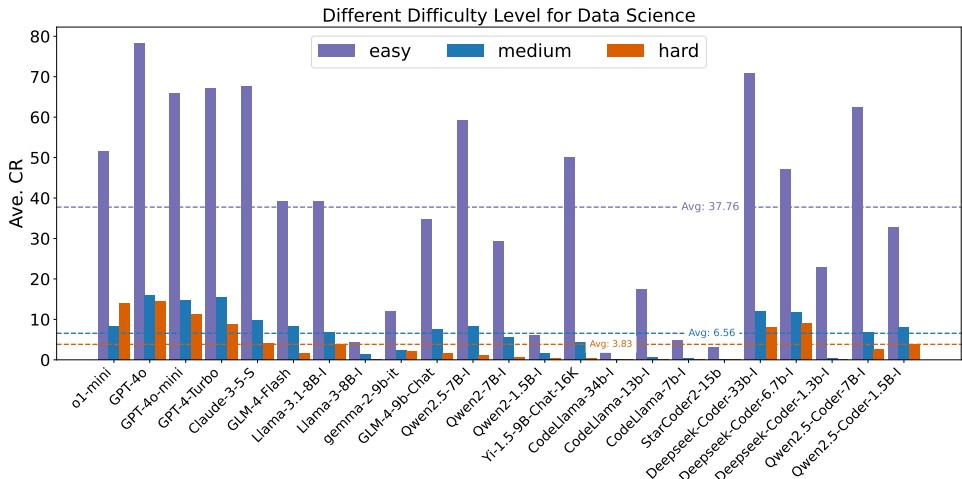

Figure 4: Average Completion Rate results regarding different difficulty levels.

attaining total scores of 47.67% and 45.99%, respectively. In contrast, there are also a few models that only pass very few tasks, achieving total scores even lower than 5.0%. Of these, CodeLlama-34B-Instruct unexpectedly achieves a score of 1.33%, even lagging behind its small-scale version CodeLlama-7B-Instruct. We present an analysis of the anomaly in Section 5.4. **(3) Furthermore, we display ranked overall scores and average scores in Figure 3. It can be concluded that API-access models basically outperform open-sourced models on average**, reaching an average score of 49.15% to 23.01% for open-access models. In comparison, the performance variance between API-based models is smaller than that of open-sourced models.

## 5.2 ABLATION STUDY ON DIFFERENT DIFFICULTY LEVELS

To evaluate multiple LLMs on their ability to complete prompts of varying difficulty, we categorized tasks using BCB and data formatted in CSV, human handwritten prompts, and data science-related DL tasks as easy - 167, medium - 30, and hard levels - 25, respectively. We assessed multiple LLMs by combining different difficulty levels, overall average CR, and the average CR for each difficulty level. From the Figure 4, it can be observed that: **(1) Consistency Across Difficulties:** Some LLMs, like GPT-4o, GPT-4o-mini, GPT-4-Turbo, and Deepseek-Coder-33B-Instruct, exhibit consistent performance across all difficulty levels, indicating robustness. **(2) Top Performers on Hard Level:** Models such as GPT-4 series and Deepseek-Coder-Instruct series are among the top performers, scoring high average CRs, particularly excelling in complex, data-driven tasks defined as hard. **(3) Performance Gaps:** There are noticeable gaps in the average CRs among general models and small-scale models, with some achieving lower scores overall, suggesting that general models are less efficient or accurate in data science tasks.

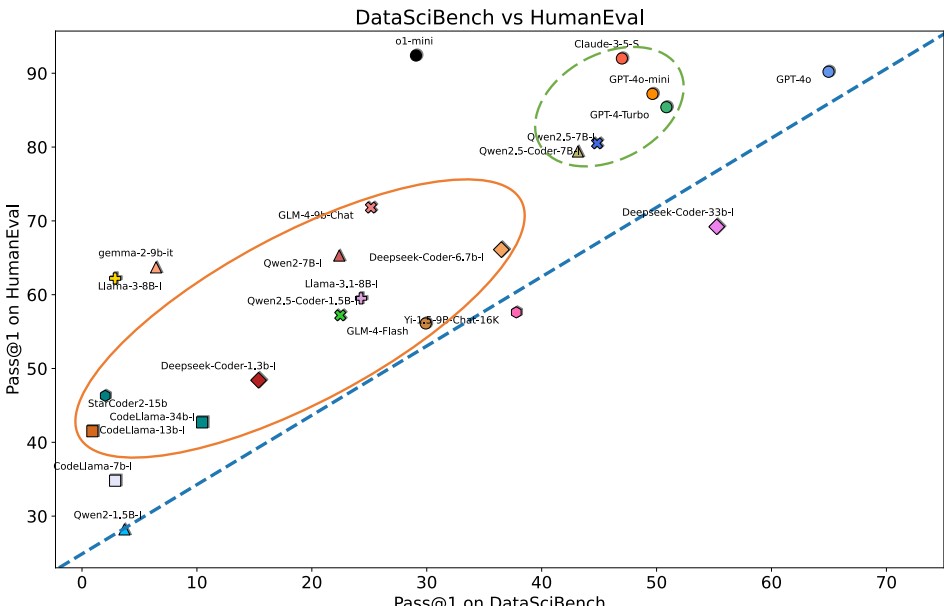

Figure 5: Pass@1 comparison of all tested LLMs between `DataSciBench` and HumanEval. Circle markers denote the API-based models while other markers denote various open-sourced LLMs. The green dashed areas indicate the LLMs perform well on the two benchmarks and the orange solid areas indicate performances of the two datasets are relatively mismatched.

## 5.3 CONTAMINATION WITH OTHER BENCHMARKS

**Comparison with HumanEval in Figure 5.** We compare our proposed `DataSciBench` with HumanEval. As shown in Figure 5, we observe that most LLMs are located in the upper triangular region of the graph and all tested models are divided into two groups, in which the green-dashed-line areas where LLMs perform well on the two benchmarks and the orange-solid-line area where performances on the two datasets with the same model indicates significant discrepancies.

**Correlation analysis with other benchmarks in Table 3.** We perform correlation analysis to evaluate the alignment between our benchmark and coding evaluations like BigCodeBench and LiveCodeBench. To achieve this, we calculate both Pearson's $r$ and Spearman's $p$ correlation coefficients, which provide insights into the strength and direction of relationships between our benchmark and these established metrics.

This analysis not only validates our results but also ensures robustness across different evaluation dimensions. Our findings indicate strong positive correlations, suggesting that our benchmark aligns well with these established coding evaluation metrics.

Table 3: Correlation

| | DataSciBench | |
|---|---|---|
| | $r$ | $p$ |
| LiveCodeBench | 0.853 | 0.673 |
| BigCodeBench | 0.823 | 0.808 |

## 5.4 INSIGHTS

With curated metrics, we are able to obtain deeper insights into LLMs' ability to plan and execute complex data science tasks. The experiment results also raise questions that are worth exploring since some results do not conform with conventional perceptions.

**Models excel at reasoning but do not necessarily perform better on complex data science tasks.** Although it's true that data science coding tasks often involve scheduling and step-by-step execution similar to reasoning scenarios, results show that even the LLMs proficient in reasoning tasks can still fail to complete complex data science tasks. For instance, the OpenAI's o1-mini model, which is commonly regarded as one of the best reasoning models, unexpectedly failed on many of `DataSciBench`'s tasks. The model only achieves a 29.77% overall success rate, significantly

lagging behind the company's previously introduced models like GPT-4o and GPT-4-Turbo. After examining the completions generated by o1-mini, we discovered that the failures are primarily caused by non-compliance with instructions, incorrect calls, and forgetfulness. While successfully splitting the task into multiple subtasks, the model often forgets to export required execution outcomes or just outputs undesired data. In other cases, the model may falsely call a library function or method that sometimes does not even exist. These facts remind us that real-life data science coding tasks often comprehensively challenge the model's ability to follow fine-grained instructions, utilize existing tools (libraries, APIs...), and do planning. To perform and align well on these tasks, a model has to be competitive on all related aspects.

**Large scale models sometimes may fail to follow simple instructions more frequently.** StarCoder2-15B performs worse than some smaller models, and CodeLlama-34B-Instruct even performs worse than its 13b and 7b versions. The main reason is that the larger-scale version lacks some other ability like generating formatted text according to prompts. Perhaps a large amount of data in a certain format is being used to train a larger version that fails to follow the prompt to generate another format different from that. Some examples can be seen in the Appendix A.15. Indeed, the larger scale model of CodeLlama also fails to outperform the smaller scale version in LCB.

# 6 RELATED WORKS

## 6.1 LLMs FOR DATA SCIENCE

With the popularity of large-scale language models, researchers have developed a series of LLM-based agents for data science. Specifically, SheetCopilot (Li et al., 2024) designs a tabular agent, which directly processes natural language-described tasks, and generates and executes a series of operation plans on datasheets to produce the desired results. Data Copilot (Zhang et al., 2024d) is an intelligent agent that serves as a bridge between users and data, which automatically executes data processing, prediction, and visualization tasks based on users' data needs. InsightPilot (Ma et al., 2023) focuses on exploratory data analysis and can automatically discover data insights related to fuzzy questions raised by users. Data interpreter (Hong et al., 2024) augments problem-solving in data science with dynamic planning with hierarchical graph structures, tool integration, and logical inconsistency identification in feedback. The correctness of data analysis in data science has a significant impact on decision-making. Therefore, with the continuous increase of data science agents, it is urgent to conduct a comprehensive and in-depth evaluation of data science agents.

## 6.2 LLM AGENT EVALUATION BENCHMARKS FOR DATA SCIENCE

Assessing the effectiveness of LLMs in handling diverse and challenging data science prompts is essential to push the boundaries of benchmarking practices in the field of data science and LLM research. Data science agents often solve problems by generating code, so the capabilities of data science agents are closely related to the code generation capabilities of large models. There are already many benchmarks for evaluating the code capability of large models. MLAgent-Bench (Huang et al., 2023) benchmarks the LLMs' abilities on traditional machine learning tasks. NaturalCodeBench (Zhang et al., 2024c) evaluates the capabilities of code generation models on the real prompts from the CodeGeeX (Zheng et al., 2023) platform. However, the general code evaluation benchmark ignores the characteristics of data science tasks and cannot comprehensively and effectively evaluate the capabilities of large models in data science.

# 7 CONCLUSION

This paper introduces `DataSciBench`, a novel framework tailored to assess the capabilities of Large Language Models (LLMs) in data science tasks. By meticulously curating challenging prompts and leveraging robust LLMs alongside a self-consistency strategy, we generate ground truth for each prompt. To comprehensively evaluate LLM performance, we aggregate evaluation metrics and synthesize the Task-Function-Code (TFC) list programmatically. Subsequently, we evaluate 23 API-based and open-source models, offer valuable research and engineering insights, and present error analyses of the assessed LLMs.

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

# A APPENDIX

## A.1 LIMITATIONS

In certain visualization tasks, our initial metrics and evaluation methods (e.g., VLM-as-a-judge) may lack precision. Further refinement of metrics is required to evaluate data visualization tasks effectively. One potential approach could involve employing Vision Language Models (VLMs) to train critic models, enhancing the capability for fine-grained evaluations of visualizations.

## A.2 COMPARISON WITH EXISTING BENCHMARKS

While DataSciBench does show a correlation with LCB or BCB in Section 5.3, our benchmark offers several unique and important contributions:

- **Domain-Specific Focus:** DataSciBench specifically targets data science and analytics tasks. However, existing benchmarks primarily focus on general programming problems. This specialization helps evaluate models' capabilities in handling real-world data analysis scenarios.
- **Task Diversity:** Our benchmark includes unique task types like data preprocessing, visualization, and statistical analysis. These tasks are underrepresented in current benchmarks. This provides deeper insights into models' data science-specific capabilities.
- **Complementary Insights:** While overall correlations exist, we observe meaningful differences in model rankings. For example, models like Meta-Llama-3-8B-Instruct and CodeLlama-34B-Instruct show distinct performance patterns. These differences highlight capabilities specific to data science tasks that other benchmarks may not capture.

The correlation with existing benchmarks validates our evaluation methodology, while our domain-specific focus provides valuable new insights for assessing AI models in data science applications.

## A.3 MOTIVETION AND EXAMPLE OF TASK-FUNCTION-CODE (TFC)

The TFC framework was developed to address several critical challenges in automated evaluation of data science tasks:

- **Systematic Task Selection:** TFC provides a structured approach to identify and categorize key tasks across six established types. This systematic organization ensures comprehensive coverage of essential data science operations and helps maintain evaluation consistency and completeness.
- **Standardized Evaluation Metrics:** Data science tasks often lack standardized evaluation criteria. TFC addresses this by explicitly defining appropriate evaluation functions for each task. For example, data preprocessing tasks require specific metrics that differ from visualization tasks. This standardization ensures fair and consistent assessment.
- **Automated Execution Framework:** TFC includes executable code components for both tasks and evaluation metrics. This automation significantly improves evaluation efficiency, result reproducibility, and testing scalability.
- **Ground Truth Generation:** TFC serves as a crucial foundation for establishing ground truth, particularly valuable for complex tasks where ground truth is not readily available, and enables systematic verification and validation of model outputs.

Overall, the TFC structure represents a novel contribution by providing a comprehensive framework that bridges the gap between task definition, evaluation criteria, and automated assessment in data science contexts.

## A.4 CAVEATS WHEN USING LLMS FOR DATA SCIENCE

Here we list the issues that occurred during testcase generation, most of which have been addressed by modifying the prompts. We notice that some of the issues may be disruptive to the system and some may be subtle but important.

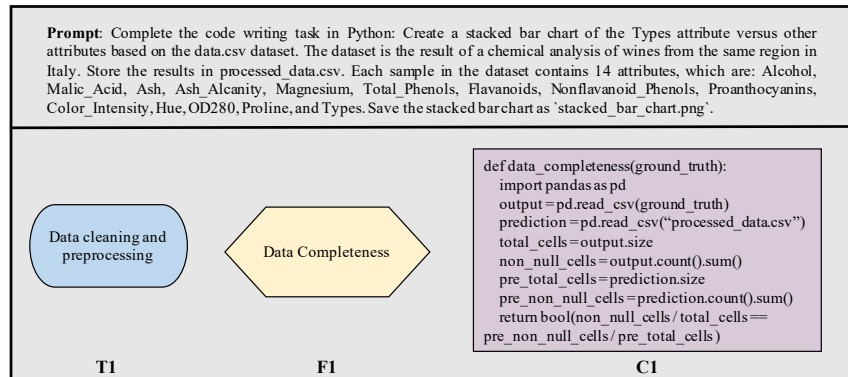

Figure 6: An example of TFC tuples.

1. Be careful when using LLMs on well-known open source datasets, especially with customized tasks and data split. LLMs may memorize some open-source datasets. For example, if we want to use part of the penguin dataset that does not contain certain columns, the model (GPT-4o) will still explicitly process those columns in the code.

2. Hallucination during data pre-processing. For example, when the model is required to merge two CSV, it may hallucinate on a common column and not go through all the columns in the files to find the actual ones.

3. On multilingual tasks. LLMs may not be able to select the correct encoding. For instance, when they are required to open a CSV file that has content in Chinese, they will struggle to choose the correct encoding to open the file. Even if they are hinted that the file is in Chinese, they may choose encodings other than "gbk", e.g., "latin".

A.5    VLM-AS-A-JUDGE PROMPT AND EXAMPLES

In this section, we define the criteria to measure quality of LLM-generated figures/charts.

**Prompt for VLM-as-a-judge**

Above are two figures, which are A and B. The first figure is the ground truth image and the second figure is the predicted image. The total score is 5. Please score B following the criteria below:

- add 1 point for Data Representation Consistency: Ensure that the underlying data represented by the two charts is identical. This includes the values for all data points and the range of the data. Any variation in the dataset used would make the charts different.
- add 1 point for Axis Labels and Scales: Verify that both charts have identical axis labels, units, and scales. Any difference in how the axes are labeled or scaled, such as using logarithmic vs. linear scales, can affect the interpretation of the data.
- add 1 point for Graphical Elements: Check if the visual elements (such as lines, bars, markers, etc.) are represented the same way in both charts. Line thickness, marker styles, and colors should match across charts for them to be considered visually equal.
- add 1 point for Legend and Annotations: Confirm that any legends, titles, or annotations (e.g., text labels, arrows, or highlights) are the same in both charts. These elements often provide crucial context for interpreting the chart.
- add 1 point for Chart Dimensions and Layout: Ensure that the dimensions (height and width), aspect ratios, and layout of the charts are identical. Even if the content and representation are similar, a different aspect ratio or spacing between elements can change the chart's overall appearance and interpretation.

Please write down the total score for B based on the criteria above, and provide a brief explanation of your reasoning. If you believe that the two figures are not identical, please explain the differences you observed.

### Explanation:
your explanation here

### Total Score:
x/5

## Example #1 of VLM-as-a-judge (Model: claude-3-5-sonnet-20240620)

Image A:

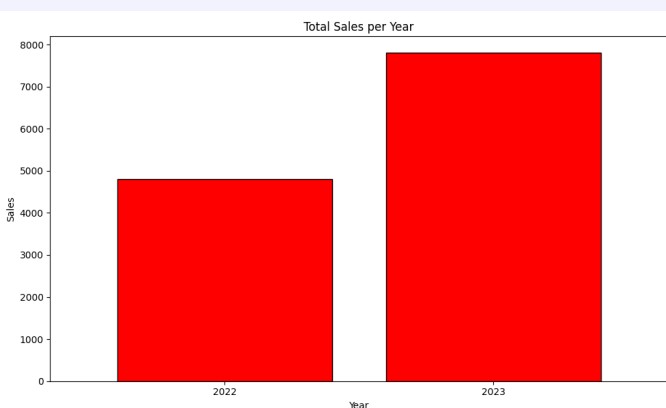

Image B:

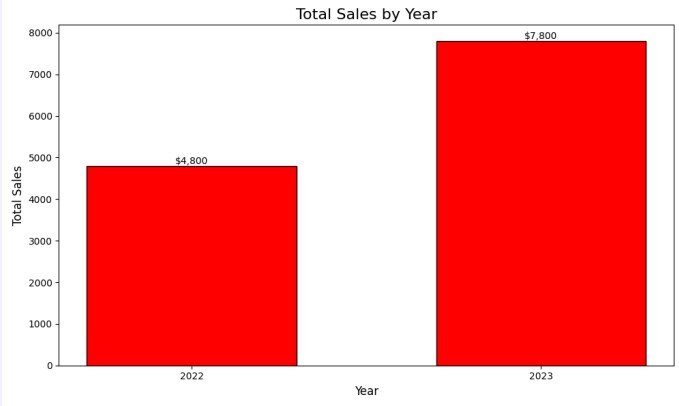

[Prompt for VLM-as-a-judge]
### Explanation:
1. **Data Representation Consistency**: Both charts represent the same underlying data for the years 2022 and 2023, with sales figures of $4,800 for 2022 and $7,800 for 2023. **+1 point**.

2. **Axis Labels and Scales**: Both charts have identical X-axis labels (years) and Y-axis labels (total sales). They both use the same scale on the Y-axis, ranging from 0 to 8000. **+1 point**.

3. **Graphical Elements**: The bars in both charts are red with the same thickness. Thus, the visual representation matches. **+1 point**.

4. **Legend and Annotations**: The second chart includes dollar amounts on each bar, which isn't present in the first chart. This difference impacts the context provided for interpreting the chart, reducing consistency. **0 points**.

5. **Chart Dimensions and Layout**: Both charts appear similar in terms of layout and dimensions, with no significant visual differences observed. **+1 point**.

Given the above evaluations, the total score for B is calculated as follows:

### Total Score:
4/5

---

**Example #2 of VLM-as-a-judge (Model: CodeLlama-13B-Instruct)**

Image A:

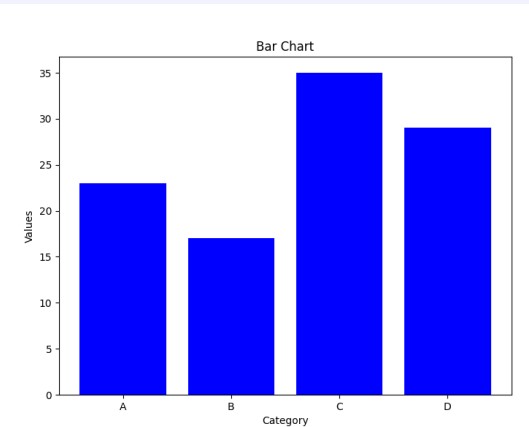

Image B:

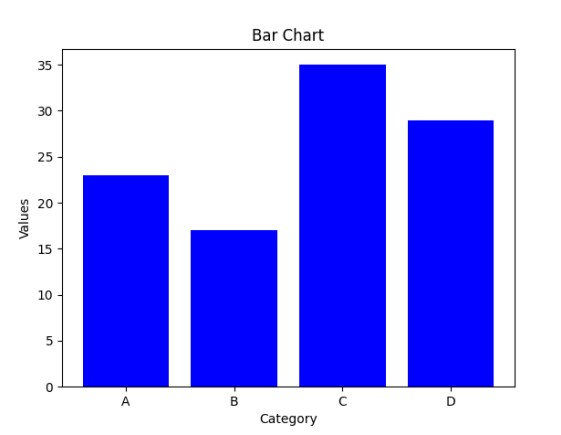

[Prompt for VLM-as-a-judge]
### Explanation:
1. **Data Representation Consistency**: Both figures (A and B) represent the same underlying data with identical values for categories A, B, C, and D. Therefore, 1 point is awarded.
2. **Axis Labels and Scales**: The axis labels ("Category" for the x-axis and "Values" for the y-axis) appear to be identical in both images, and the scale for the y-axis seems consistent. Thus, 1 point is awarded.
3. **Graphical Elements**: Both charts use blue bars to represent the data, and the thickness and style of the bars are similar. Therefore, 1 point is awarded.
4. **Legend and Annotations**: There are no legends or additional annotations in either image, and the titles are the same ("Bar Chart"). Thus, 1 point is awarded.
5. **Chart Dimensions and Layout**: The dimensions and overall layout of the two charts differ. Image A has a different aspect ratio compared to Image B, which affects the total layout. Hence, 0 points are awarded.

### Total Score:
4/5

**Example #3 of VLM-as-a-judge (Model: o1-mini)**

Image A:

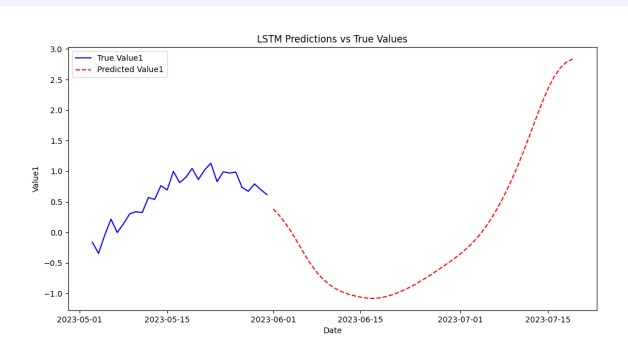

Image B:

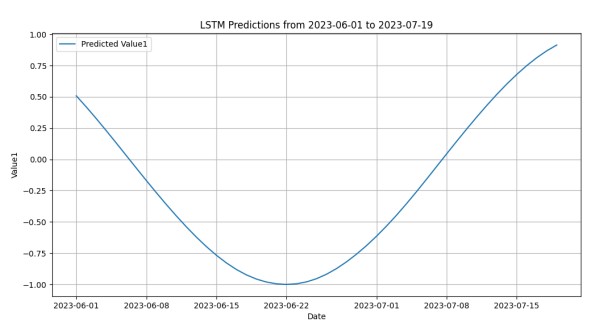

[Prompt for VLM-as-a-judge]
### Explanation:
1. **Data Representation Consistency**: The two figures represent different sets of data. Image A shows the true values compared to the predictions over a broader date range, while Image B only displays the predicted values for a specific date range (from 2023-06-01 to 2023-07-19). This difference in data sets indicates that the underlying data represented is not identical. **(Score: 0)**

2. **Axis Labels and Scales**: Both images have similar axis labels ("Date" for the x-axis and "Value1" for the y-axis), but Image B lacks clarity on the exact date range it represents, while Image A has a wider range and different intervals. Thus, while the labels seem consistent, the scale differs due to the limited timeframe in Image B. **(Score: 0)**

3. **Graphical Elements**: Image A uses a dashed red line for the predicted values, while Image B uses a solid blue line for predictions. This difference in line style and color affects visual representation. **(Score: 0)**

4. **Legend and Annotations**: Image A includes a legend identifying its two lines distinctly, while Image B only has a legend for the predicted values. The titles are also different. This inconsistency affects how the charts convey context and interpretability. **(Score: 0)**

5. **Chart Dimensions and Layout**: The overall dimensions and aspect ratios of the charts are visually different. Image A has a wider aspect ratio, accommodating a broader date range, while Image B has a more condensed look. **(Score: 0)**

### Total Score:
0/5

## A.6 RESULTS

We present the detailed results for our collected prompt and BigCodeBench source in Table 4 and Table 5.

Table 4: Evaluation results for `DataSciBench` (our collected prompts).

| Models | Size | Coarse-grained Metrics | | Fine-grained Metrics | | | | | | Score |
|---|---|---|---|---|---|---|---|---|---|---|
| | | Success (%) | CR (%) | VLM | F1 | F2 | F3 | F4 | F5 | |
| o1-mini | N/A | 13.45 | 15.43 | 1.75 | 53.75 | 0.00 | 45.50 | 46.25 | 25.79 | 19.35 |
| GPT-4o-2024-05-13 | N/A | **19.82** | **17.89** | 2.10 | **71.79** | 0.00 | **54.50** | 70.00 | 33.95 | **24.24** |
| GPT-4o-mini | N/A | 12.73 | 17.35 | 1.65 | 62.86 | **15.00** | 50.50 | 69.38 | **43.68** | 24.07 |
| GPT-4-Turbo | N/A | 17.27 | 17.36 | 1.85 | 65.18 | 10.00 | 44.50 | **78.12** | 41.05 | 24.18 |
| Claude-3-5-Sonnet-20240620 | N/A | 8.00 | 11.12 | 1.44 | 32.32 | 0.00 | 36.00 | 63.12 | 17.63 | 15.15 |
| GLM-4-Flash | N/A | 9.82 | 7.43 | 1.51 | 47.14 | 0.00 | 22.00 | 24.38 | 3.95 | 10.27 |
| Meta-Llama-3.1-8B-Instruct | 8B | 10.00 | 7.72 | 1.55 | 45.18 | 0.00 | 16.00 | 23.12 | 8.95 | 10.26 |
| Meta-Llama-3-8B-Instruct | 8B | 1.64 | 1.43 | 1.93 | 7.86 | 0.00 | 4.50 | 6.88 | 0.00 | 2.07 |
| Gemma-2-9B-it | 9B | 5.64 | 5.51 | 1.63 | 26.79 | 0.00 | 13.00 | 22.50 | 2.89 | 7.20 |
| GLM-4-9B-Chat | 9B | 10.55 | 9.96 | 1.56 | 55.36 | 0.00 | 31.00 | 28.75 | 21.32 | 13.90 |
| Qwen2.5-7B-Instruct | 7B | 11.64 | 10.11 | 1.44 | 55.36 | 0.00 | 36.50 | 33.12 | 18.42 | 14.40 |
| Qwen2-7B-Instruct | 7B | 6.91 | 5.90 | 1.68 | 32.50 | 0.00 | 18.00 | 21.88 | 2.37 | 8.00 |
| Qwen2-1.5B-Instruct | 1.5B | 1.82 | 1.60 | 1.54 | 3.57 | 0.00 | 2.00 | 13.12 | 0.79 | 2.18 |
| Yi-1.5-9B-Chat-16K | 9B | 6.18 | 4.25 | 1.82 | 30.36 | 0.00 | 16.00 | 8.75 | 3.95 | 6.12 |
| CodeLlama-34B-Instruct | 34B | 0.00 | 0.03 | 0.00 | 0.00 | 0.00 | 0.00 | 0.00 | 0.00 | 0.02 |
| CodeLlama-13B-Instruct | 13B | 0.73 | 0.50 | **4.00** | 4.46 | 0.00 | 0.00 | 3.75 | 0.00 | 0.97 |
| CodeLlama-7B-Instruct | 7B | 0.55 | 0.27 | 0.00 | 1.96 | 0.00 | 0.00 | 0.00 | 0.00 | 0.30 |
| StarCoder2-15B | 15B | 0.18 | 0.20 | 2.33 | 0.54 | 0.00 | 0.00 | 0.62 | 0.00 | 0.31 |
| Deepseek-Coder-33B-instruct | 33B | 12.55 | 13.53 | 1.73 | 62.86 | 0.00 | 43.00 | 51.88 | 21.32 | 18.46 |
| Deepseek-Coder-6.7B-instruct | 6.7B | 12.55 | 13.56 | 1.30 | 63.21 | 0.00 | 39.00 | 53.75 | 21.05 | 18.36 |
| Deepseek-Coder-1.3B-instruct | 1.3B | 0.73 | 0.61 | 3.33 | 3.39 | 0.00 | 0.00 | 1.25 | 0.00 | 0.83 |
| Qwen2.5-Coder-7B-Instruct | 7B | 6.18 | 7.87 | 1.35 | 40.18 | 0.00 | 27.50 | 33.75 | 4.47 | 10.79 |
| Qwen2.5-Coder-1.5B-Instruct | 1.5B | 6.18 | 7.52 | 1.11 | 38.57 | 0.00 | 15.50 | 40.00 | 10.53 | 10.48 |

Table 5: Evaluation results for `DataSciBench` (BigCodeBench source).

| Models | Size | Coarse-grained Metrics | | Fine-grained Metrics | | | | | | Score |
|---|---|---|---|---|---|---|---|---|---|---|
| | | Success (%) | CR (%) | F1 | F2 | F3 | F4 | F5 | | |
| o1-mini | N/A | 35.15 | 55.08 | 41.62 | 25.62 | 32.89 | 25.90 | 23.16 | | 47.77 |
| GPT-4o-2024-05-13 | N/A | **81.62** | **85.09** | **77.30** | **74.63** | **74.21** | **71.79** | **65.48** | | **81.81** |
| GPT-4o-mini | N/A | 63.11 | 71.10 | 59.46 | 58.89 | 60.26 | 55.90 | 56.79 | | 67.49 |
| GPT-4-Turbo | N/A | 63.35 | 72.54 | 61.35 | 52.04 | 62.11 | 54.36 | 53.95 | | 68.14 |
| Claude-3-5-Sonnet-20240620 | N/A | 60.48 | 73.59 | 54.59 | 49.11 | 62.37 | 49.49 | 55.39 | | 68.08 |
| GLM-4-Flash | N/A | 37.07 | 42.8 | 33.04 | 39.11 | 36.05 | 28.72 | 17.89 | | 39.55 |
| Meta-Llama-3.1-8B-Instruct | 8B | 29.58 | 42.51 | 35.95 | 24.26 | 23.95 | 22.82 | 31.41 | | 38.16 |
| Meta-Llama-3-8B-Instruct | 8B | 3.29 | 4.74 | 2.97 | 1.67 | 2.11 | 1.28 | 1.96 | | 3.98 |
| Gemma-2-9B-it | 9B | 7.54 | 12.81 | 25.95 | 22.46 | 27.37 | 16.67 | 21.84 | | 15.06 |
| GLM-4-9B-Chat | 9B | 30.72 | 37.11 | 23.65 | 30.78 | 27.11 | 26.67 | 18.42 | | 33.84 |
| Qwen2.5-7B-Instruct | 7B | 54.43 | 64.12 | 49.80 | 48.40 | 50.79 | 49.23 | 40.15 | | 59.52 |
| Qwen2-7B-Instruct | 7B | 28.08 | 32.06 | 30.41 | 27.63 | 32.26 | 27.18 | 9.21 | | 30.18 |
| Qwen2-1.5B-Instruct | 1.5B | 4.67 | 6.73 | 4.86 | 2.63 | 3.68 | 3.33 | 6.00 | | 5.97 |
| Yi-1.5-9B-Chat-16K | 9B | 48.74 | 54.9 | 40.70 | 48.34 | 42.11 | 46.41 | 35.64 | | 51.53 |
| CodeLlama-34B-Instruct | 34B | 1.20 | 1.94 | 1.35 | 1.11 | 2.63 | 2.05 | 1.58 | | 1.85 |
| CodeLlama-13B-Instruct | 13B | 13.71 | 19.3 | 14.05 | 15.07 | 12.53 | 17.95 | 6.84 | | 17.52 |
| CodeLlama-7B-Instruct | 7B | 3.65 | 5.19 | 4.05 | 3.15 | 3.42 | 2.31 | 2.11 | | 4.57 |
| StarCoder2-15B | 15B | 2.69 | 3.41 | 3.24 | 2.41 | 2.11 | 4.36 | 1.58 | | 3.21 |
| Deepseek-Coder-33B-instruct | 33B | 70.12 | 76.94 | 66.58 | 62.63 | 63.16 | 64.87 | 57.59 | | 73.11 |
| Deepseek-Coder-6.7B-instruct | 6.7B | 45.09 | 50.86 | 37.00 | 45.96 | 48.79 | 44.10 | 17.11 | | 47.50 |
| Deepseek-Coder-1.3B-instruct | 1.3B | 20.36 | 25.05 | 16.22 | 19.44 | 17.63 | 21.28 | 10.53 | | 22.81 |
| Qwen2.5-Coder-7B-Instruct | 7B | 58.02 | 68.01 | 55.34 | 57.44 | 49.26 | 45.38 | 45.36 | | 63.15 |
| Qwen2.5-Coder-1.5B-Instruct | 1.5B | 28.20 | 35.60 | 26.94 | 28.96 | 26.74 | 26.15 | 18.32 | | 32.69 |

## A.7 PROGRAMMATIC RULES

Table 6: Details of programmatic rules.

| Aggregate Function | Task | Type | Rule | Comparison | GT | Threshold |
|---|---|---|---|---|---|---|
| Mean Squared Error | Calculate MSE | Decimal | Bool | It $\leq$ to GT is 1, it > GT is 0 | Yes | - |
| Data Cleaning Completeness | row/column number | Integer | Bool | If it == GT, it is 1; if it != GT, it is 0 | Yes | - |
| Silhouette Score | Calculate | Decimal | Bool | If it $\geq$ to GT, it is 1, if it $\leq$ 0 | Yes | 0.5 |
| Model Accuracy | Calculate F1 | Decimal | Bool | If it $\geq$ to GT, it is 1, if it $\leq$ 0 | Yes | 0.9 |
| Model Accuracy | Calculate Accuracy | Decimal | Bool | If it $\geq$ to GT, it is 1, if it $\leq$ 0 | Yes | 0.9 |
| R-squared Value | Calculate | Decimal | Bool | If it $\geq$ to GT, it is 1, if it $\leq$ 0 | Yes | 0.9 |
| Data Cleaning Completeness | number of nulls | Integer | Bool | If it == 0, it returns 1; if it > 0, it returns 0 | Yes | - |
| Data Quality Score | number of outlier | Integer | Bool | If it == GT, it is 1; if it != GT, it is 0 | Yes | - |
| Data Accuracy | MSE | Decimal | Bool | Threshold judgment | Yes | 0.05 |
| Data Completeness | Null number ratio | Decimal | Bool | If it == GT, it is 1; if it != GT, it is 0 | Yes | - |
| Data Quality Score | Calculateloss | Decimal | Bool | It $\leq$ to GT is 1, it > GT is 0 | Yes | - |
| Association Rule Confidence | Association rule accuracy | Decimal | Bool | Threshold judgment | Yes | 0.9 |

## A.8 RELATED WORK IN DATA SCIENCE

Recently, some evaluation benchmarks for large language models in data science have been proposed. Text2Analysis (He et al., 2023) constructs the evaluation benchmark to evaluate the model's ability to handle data analysis functions and fuzzy questions on tabular data. Their prompts are obtained through manual annotation and large model generation. Furthermore, DAEval (Hu et al., 2024) is developed as another evaluation benchmark and it contains 257 data analysis questions on CSV data and questions, which are generated by LLMs. However, the prompts in these two works often only involve one task, and these prompts involve relatively simple data analysis operations. In practical data science analysis tasks, user questions often involve multiple tasks and involve performing complex data analysis operations. Therefore, we aim to provide a data science evaluation benchmark that is more in line with practical scenarios, especially for problems involving multiple subtasks and complex data analysis operations.

## A.9 QUALIFIED PROMPTS

- **Original Prompt 1:**
  There is a dataset with missing values in a CSV file, which records the region, height, weight, age, and salary of 36 individuals. Please address the following issues:
  (a) Calculate the proportion of missing values in each column and select the rows with at least two non-missing values in the last three columns.
  (b) Please fill in the weight column reasonably by combining the data from the height and region columns.

- **Qualified Prompt 1:**

  > **Qualified Prompt 1:**
  >
  > Given a dataset with missing values in a file named 'data.csv' which records the region, height, weight, age, and salary of 36 individuals, please address the following issues:
  > (a) Calculate the proportion of missing values in each column and select the rows with at least two non-missing values in the last three columns. Save your output in a CSV file named 'missing_values_proportion.csv'.
  > (b) Fill in the weight column reasonably by combining the data from the height and region columns. Save this updated dataset in a CSV file named 'updated_data.csv'.

- **Original Prompt 2:** You are required to analyze and visualize the "Global Terrorism Database" from Kaggle. Please load the dataset and perform data cleaning by handling missing values, removing duplicates, and correcting any anomalies. Conduct an exploratory data analysis (EDA) to understand the distribution and relationships within the dataset. Calculate basic statistical indicators such as mean, median, standard deviation, and provide summary statistics for key features like attack type, target type, and region.

Generate visualizations to uncover patterns and insights. Create histograms and box plots to display the distribution of numerical features, and bar plots to show the frequency of categorical variables. Use scatter plots and heatmaps to visualize relationships and correlations between features. Identify patterns in the data related to terrorist activities. For instance, determine trends over time, geographical hotspots, and common attack methods. Use clustering techniques ( K-means clustering) to identify patterns and group similar incidents together.

- **Qualified Prompt 2:**

**Qualified Prompt 2:**

You are required to analyze and visualize the *Global Terrorism Database* from Kaggle. Please follow the steps below:

1. Load the dataset
   Input: `gtd.csv`
   Output: `loaded_data.csv` (This should contain the original data loaded without any modifications.)

2. Data Cleaning
   - Handle missing values
   - Remove duplicates
   - Correct anomalies

   Input: `loaded_data.csv`
   Output: `cleaned_data.csv` (This should reflect the cleaned dataset, ready for analysis.)

3. Exploratory Data Analysis (EDA)
   - Calculate basic statistical indicators such as mean, median, and standard deviation
   - Provide summary statistics for key features (attack type, target type, region)

   Input: `cleaned_data.csv`
   Output: `eda_summary_statistics.csv` (This should include all calculated statistics for key features.)

4. Generate Visualizations
   - Create histograms and box plots for numerical features
   - Generate bar plots for categorical variables
   - Use scatter plots and heatmaps to visualize relationships and correlations

   Input: `cleaned_data.csv`
   Output: `visualizations.pdf` (This should include all visualizations generated in a single PDF file.)

5. Identify Patterns in Data Related to Terrorist Activities
   - Determine trends over time
   - Identify geographical hotspots
   - Analyze common attack methods

   Input: `cleaned_data.csv`
   Output: `patterns_analysis.csv` (This should summarize the identified patterns, trends, and hotspots.)

6. Clustering Techniques
   - Use K-means clustering to identify patterns and group similar incidents

   Input: `cleaned_data.csv`
   Output: `clustering_results.csv` (This should include the results of the clustering analysis, showing which group each incident belongs to.)

Ensure that each output file reflects the quality of the completion of the respective subtask for further evaluation.

## A.10   MODIFIED PROMPTS

- **Original Prompt 1:**

Searches a directory for CSV files matching a given regular expression pattern, reads sales data from these files, and plots the sales data with month on the x-axis and sales on the y-axis.

Note that: Each CSV file contains two columns: `Month` and `Sales`.

The function should output with:

A list of `matplotlib.axes._axes.Axes` objects, each representing a plot of sales data from a matched CSV file.

You should write self-contained code starting with:

```
import os
import pandas as pd
import re
import matplotlib.pyplot as plt
def task_func(directory: str, pattern: str) -> list:
```

• **Modified Prompt 1:**

---

**Modified Prompt 1:**

Search a directory for CSV files matching a given regular expression pattern, read sales data from these files, and plot the sales data with month on the x-axis and sales on the y-axis.

Input Requirements:

- Input Directory: `data`.
- Input Pattern: `"csv_\d+\.csv"`.

Output Requirements:

1. A list of `matplotlib.axes._axes.Axes` objects representing the plot of sales data from each matched CSV file.

2. Save each plot as a separate image file:
    - File format: PNG
    - Output filenames: `"sales_plot_<filename>.png"` where `<filename>` is the name of the CSV file without the extension.

Input File Specification:

- Each CSV file should contain two columns: `'Month'` and `'Sales'`. The input files will be located in the specified directory.

You should write self-contained code starting with:

```
import os
import pandas as pd
import re
import matplotlib.pyplot as plt

def task_func(directory: str, pattern: str) -> list:
```

---

- **Original Prompt 2:**

  Plot a scatter graph of tuples and highlight the tuple with the maximum value at index 1.
  The function should output with:
  `matplotlib.axes.Axes`: The Axes object of the plot for further manipulation and testing, with the title 'Max Tuple Highlighted', x-axis labeled 'x', y-axis labeled 'y', and a legend.
  You should write self-contained code starting with:

  ```
  import numpy as np
  from operator import itemgetter
  import matplotlib.pyplot as plt
  def task_func(data):
  ```

- **Modified Prompt 2:**

  > **Modified Prompt 2:**
  >
  > Plot a scatter graph of tuples and highlight the tuple with the maximum value at index 1 using the input data from `"data.csv"`. The function should output the following:
  >
  > A scatter plot saved as `"scatter_plot.png"` with the title 'Max Tuple Highlighted', x-axis labeled 'x', y-axis labeled 'y', and a legend. The highlighted point should signify the tuple with the maximum value at index 1.
  >
  > Please write self-contained code starting with:
  >
  > ```
  > import numpy as np
  > from operator import itemgetter
  > import matplotlib.pyplot as plt
  > def task_func(data):
  > ```

## A.11 PROMPT EXAMPLES OF DIFFERENT DIFFICULTY LEVELS

> **Easy-level Prompt 1:**
>
> ```
> {
> "prompt":
> ```
> "Read the dataset (input file: "Fish.csv"). Encode the dataset to divide it into training and test sets. From the dataset's four categories of Bream, Roach, Parkki, and Perch, randomly select 2 samples from each category for the test set. The remaining samples will be used as the training set. Output the number of samples in the training and test sets in a CSV file named "sample_counts.csv".
>
> Next, implement the KNN algorithm with K values of 1, 5, 15, and 100 to classify all samples in the test set. Output the classification results of the test samples to a CSV file named "classification_results.csv".",
>
> `"data_source_type"`: "2=open source data"
> `}`

## A.12 Prompt examples of different difficulty levels

---

**Easy-level Prompt 2:**

```
{
"prompt":
```
"Using the dataset of house prices and economic indicators provided below, please complete the following subtasks. Each subtask should generate an output file for evaluation, with specific input files where applicable.

Dataset: `"data.csv"`

Subtasks:

1. Gaussian Naive Bayes Training: Train a Gaussian Naive Bayes model on the cleaned dataset and evaluate its accuracy.
Input: `"data.csv"`
Output: `"gaussian_model_accuracy.csv"` (contains accuracy metrics)

2. Multinomial Naive Bayes Training: Train a Multinomial Naive Bayes model on the cleaned dataset and evaluate its accuracy.
Input: `"data.csv"`
Output: 'multinomial_model_accuracy.csv' (contains accuracy metrics)

3. Comparison of Models: Generate a comparative analysis of the accuracy of both models in a visual format.
Input: `"gaussian_model_accuracy.csv"`, `"multinomial_model_accuracy.csv"`
Output: `"model_comparison.png"`

Please ensure that the outputs contain necessary details for further evaluation.",

```
"data_source_type": "3=human written data"
}
```

---

## A.13 PROMPT EXAMPLES OF DIFFERENT DIFFICULTY LEVELS

---

**Medium-level Prompt:**

```
{
"prompt":
```
"Given a DataFrame `df` with sales data containing columns: TransactionID, ProductID, Quantity, SaleDate, StoreID, Revenue:

1. Handle missing Revenue values: Impute any missing Revenue values with the median of the Revenue column.
Input file: `data.csv`
Output: `step1_imputed_revenue.csv` (contains the DataFrame after handling missing values)

2. Detect and replace outliers: Detect and replace outliers in Quantity and Revenue. Outliers are defined as values below the 1st percentile or above the 99th percentile. Replace them with the corresponding 1st or 99th percentile value instead of removing them.
Input file: `step1_imputed_revenue.csv`
Output: `step2_replaced_outliers.csv` (contains the DataFrame after outlier replacement)

3. Normalize Quantity and Revenue: Normalize Quantity and Revenue using Z-score normalization.
Input file: `step2_replaced_outliers.csv`
Output: `step3_normalized_data.csv` (contains the DataFrame after normalization)

4. Ensure SaleDate format: Ensure SaleDate is in datetime format.
Input file: `step3_normalized_data.csv`
Output: `step4_formatted_dates.csv` (contains the DataFrame after ensuring datetime format)

5. Encode ProductID and StoreID: Encode the ProductID and StoreID columns using separate label encoders to avoid any potential overlap in numerical values between categories from different columns.
Input file: `step4_formatted_dates.csv`
Output: `final_cleaned_data.csv` (contains the final cleaned DataFrame)

Perform the specified data cleaning and preprocessing tasks and output the cleaned DataFrame as the final result.",

```
"data_source_type": "3=human written data"
}
```

1404
1405

## A.14    PROMPT EXAMPLES OF DIFFERENT DIFFICULTY LEVELS

1406
1407

**Hard-level Prompt:**

1408
1409
1410
1411
1412

{ "prompt":
"Write a Graph Recurrent Neural Network (GRNN) model based on attention mechanisms using Python for processing and analyzing time series data. Ensure to meet the following requirements:

1413
1414

1. "Graph network design": Create a graph network where each graph represents an aerial formation, and the number of nodes corresponds to the number of vehicles in the formation.
- "Output": Graph structure representation file.
- "Output filename": "graph_structure.json"

1415
1416
1417
1418

2. "Data format": The time series data of aerial target formations is stored in Excel files located in the ".data/" directory, where each Excel file contains multiple sheets, with each sheet representing the time series data of a vehicle.
- "Input": Excel files from "..data/" directory.
- "Input filename": from "data_1.xlsx" to "data_5.xlsx"

1419
1420
1421
1422
1423

3. "Data reading": Read all Excel files from the "..data/" directory and extract the sheets for processing.
- "Output": Combined time series data in a structured format.
- "Output filename": "combined_data.csv"

1424
1425
1426
1427

4. "Data segmentation": Segment the time series data of each vehicle using a sliding window with length "l" and step size "s".
- "Output": Segmented time series data.
- "Output filename": "segmented_data.csv"

1428
1429
1430
1431
1432

5. "Training and testing set division": Divide the segmented data into training and testing sets with a 7:3 ratio.
- "Output": Training and testing datasets.
- "Output filenames": "train_data.csv", "test_data.csv"

1433
1434
1435
1436
1437

6. "Labels": The last column of each Excel file contains label data, which should be preserved during the reading and segmentation process.

1438
1439

7. "Model training": Train the GRNN model using the training set to identify the action intentions of aerial target formations.
- "Output": Trained model file.
- "Output filename": "trained_model.h5"

1440
1441
1442
1443
1444

8. "Training parameters": Set the number of training epochs to 60, and record the loss value, accuracy, recall, and F1-Score for each epoch.
- "Output": Training metrics log file.
- "Output filename": "training_metrics.csv"

1445
1446
1447
1448
1449

9. "Model evaluation": Evaluate the trained model with the test set and generate a classification report and confusion matrix.
- "Output": Classification report and confusion matrix visualization.
- "Output filenames": "classification_report.txt", "confusion_matrix.png"

1450
1451
1452
1453

Please provide a complete code implementation and ensure that the code structure is clear and well-commented for understanding and evaluation.",
"data_source_type": "3=human written data"
}

1454
1455
1456
1457

## A.15 EXAMPLES OF WORSE PERFORMANCE OF LARGER MODEL

- **Generated by CodeLlama-34B-Instruct in one task:**

```python
[PYTHON]
# Define the context and task
context = {
    "data": "https://s3.amazonaws.com/benbensaid-public-data/data.csv",
    "target": "y"
}

task = {
    "task_type": "data exploration",
    "instruction": "Explore the data and identify any patterns or trends."
}

# Generate the plan
plan = generate_plan(context, task)

# Print the plan
print(plan)
[/PYTHON]
```

- **Generated by StarCoder2-15B in one task:**

```
Sure, here's a plan for your context:
1. Find the videos with titles containing "how" or "what"
2. Calculate the like ratios
3. Visualize the like ratios
```

- **The expected format should at least begin and end with three backticks and must comply with JSON format. Here's an example:**

```
"
[
    {
        "task_id": "1",
        "dependent_task_ids": [],
        "instruction": "Preprocess and clean the data.",
        "task_type": "preprocessing and cleaning"
    },
    {
        "task_id": "2",
        "dependent_task_ids": ["1"],
        "instruction": "Explore the data to find patterns and insights.",
        "task_type": "data exploration"
    },
    {
        "task_id": "3",
        "dependent_task_ids": ["2"],
        "instruction": "Visualize the data to communicate insights.",
        "task_type": "data visualization"
    },
    {
        "task_id": "4",
        "dependent_task_ids": ["3"],
        "instruction": "Generate an interpretability report.",
        "task_type": "interpretability report"
    }
]
"
```

## B   ERROR ANALYSIS AND CASE STUDY

We have analyzed the common errors during problem solving, summarize them here, and then study several cases, including a successful case and a failure case. The common errors includes:

1. Coding errors when solving data science problems using codes. And based on our observation, the main kind of these is execution error. It may be due to different reasons. For example, hallucination on the column name of a csv file.

2. Json format errors. These errors come from the agent framework side, where they use json format to wrap up actions, e.g. `WriteAnalysis`.

## B.1    A Successful case

---

**A successful case from claude-3-5-sonnet-20240620**

**Prompt**: You are given a list of tickers and their daily closing prices for a given period stored in an input file named "data.csv". Implement the most_corr function that, when given the daily closing prices, returns the pair of tickers that are the most highly (linearly) correlated by daily percentage change. The function should follow these specifications:

1. **Input File**: "../data.csv" - A CSV file containing a dataframe with columns representing each ticker's daily closing prices.

2. **Output Requirements**: The function should generate the following output:
- **Output Format**: A CSV file.
- **Output Filename**: "most_corr_output.csv" - This file should contain two columns labeled "Ticker 1" and "Ticker 2", representing the pair of tickers that are most correlated by daily percentage change.

Here's the starter code for your implementation:

```python
import pandas as pd
import numpy as np

def most\_corr(prices):
    """
    :param prices: (pandas.DataFrame) A dataframe containing
    each ticker's daily closing prices.
    :returns: (container of strings) A container, containing
    the two tickers that are the most highly (linearly)
    correlated by daily percentage change.
    """
    return None

# For example, the code below should print: ('FB', 'MSFT')
print(most_corr(pd.DataFrame.from_dict({
    'GOOG' : [
        742.66, 738.40, 738.22, 741.16,
        739.98, 747.28, 746.22, 741.80,
        745.33, 741.29, 742.83, 750.50
    ],
    'FB' : [
        108.40, 107.92, 109.64, 112.22,
        109.57, 113.82, 114.03, 112.24,
        114.68, 112.92, 113.28, 115.40
    ],
    'MSFT' : [
        55.40, 54.63, 54.98, 55.88,
        54.12, 59.16, 58.14, 55.97,
        61.20, 57.14, 56.62, 59.25
    ],
    'AAPL' : [
        106.00, 104.66, 104.87, 105.69,
        104.22, 110.16, 109.84, 108.86,
        110.14, 107.66, 108.08, 109.90
    ]
})))
```

---

---

**A successful case from claude-3-5-sonnet-20240620**

**Output**: The following example successfully fulfill the requirement of the given prompt by completing the function provided in the prompt and write the function to `most_corr_output.csv`.

```
[
    {
        'task_id': '1',
        'dependent_task_ids': [],
        'instruction': "Load and preprocess the input data from
        'data.csv'",
        'task_type': 'preprocessing and cleaning',
        'code': '''
        import pandas as pd
        import numpy as np

        # Load the data from 'data.csv'
        prices = pd.read_csv('../data.csv')

        # Set the first column (assuming it's a date column) as
        the index
        prices.set_index(prices.columns[0], inplace=True)

        # Calculate daily percentage changes
        pct_changes = prices.pct_change()

        # Drop the first row (NaN values due to pct_change
        calculation)
        pct_changes = pct_changes.dropna()

        print("Data loaded and preprocessed. Shape of \
        percentage changes:", pct_changes.shape)
        print("Columns (tickers):", \
        pct_changes.columns.tolist())
        print("First few rows of percentage changes:")
        print(pct_changes.head())
        ''',
        'result': "Data loaded and preprocessed. Shape of \
        percentage changes: (29, 4)
        Columns (tickers): ['Ticker2', 'Ticker3', 'Ticker4',\
        'Ticker5']
        First few rows of percentage changes:
        Ticker2   Ticker3   Ticker4   Ticker5
        Ticker1
        65.60   -0.615289  0.108929  0.002184  0.841616
        52.06    1.633757 -0.024740 -0.353101 -0.435643
        68.34   -0.452888 -0.230862  0.308297  0.160458
        111.19  -0.204800 -0.227069 -0.070287  0.208418
        128.52   0.094136  0.280394  0.260849 -0.428407
        ",
        'is_success': True,
        'is_finished': True
    },
```

---

A successful case from claude-3-5-sonnet-20240620

```
{
    'task_id': '2',
    'dependent_task_ids': ['1'],
    'instruction': 'Implement the most_corr function to
    find the most correlated pair of tickers',
    'task_type': 'data mining',
    'code': '''
    def most_corr(prices):
        """
        :param prices: (pandas.DataFrame) A dataframe
        containing each ticker's
                    daily closing prices.
        :returns: (container of strings) A container,
        containing the two tickers that
                    are the most highly (linearly) correlated
                    by daily percentage change.
        """
        # Calculate percentage changes
        pct_changes = prices.pct_change().dropna()

        # Calculate correlation matrix
        corr_matrix = pct_changes.corr()

        # Set diagonal and lower triangle to NaN to avoid
        self-correlation and duplicates
        corr_matrix = corr_matrix.where(np.triu( \
        np.ones(corr_matrix.shape), k=1).astype(bool))

        # Find the location of the maximum correlation
        max_corr = corr_matrix.max().max()
        max_corr_idx = corr_matrix.stack().idxmax()

        # Get the tickers with the highest correlation
        ticker1, ticker2 = max_corr_idx

        # Create output DataFrame
        output_df = pd.DataFrame({'Ticker 1': [ticker1], \
        'Ticker 2': [ticker2]})

        # Save output to CSV
        output_df.to_csv('./most_corr_output.csv', \
        index=False)

        return (ticker1, ticker2)

    # Test the function with the data we've already loaded
    result = most_corr(prices)
    print(f"The most correlated pair of tickers: {result}")

    # Read and display the output CSV file
    output = pd.read_csv('./most_corr_output.csv')
    print("\nContents of most_corr_output.csv:")
    print(output)''',
    'result': "The most correlated pair of tickers: ...
    [omitted]
    'is_success': True,
    'is_finished': True
},
```

```
A successful case from claude-3-5-sonnet-20240620

    {
        'task_id': '3',
        'dependent_task_ids': ['2'],
        'instruction': "Generate and save the output CSV file
        'most_corr_output.csv'",
        'task_type': 'other',
        'code': '''
        import pandas as pd

        # Read the output CSV file
        output_df = pd.read_csv('./most_corr_output.csv')

        # Display the contents of the file
        print("Contents of most_corr_output.csv:")
        print(output_df)

        # Verify that the file contains the expected columns
        expected_columns = ['Ticker 1', 'Ticker 2']
        assert all(col in output_df.columns for col in \
        expected_columns),"Output file is missing expected \
        columns"

        # Verify that the file contains exactly one row
        assert len(output_df) == 1,"Output file should \
        contain exactly one row"

        print("Output file verification complete. The file \
        meets the expected format.")''',
        'result': 'Contents of most_corr_output.csv:
        Ticker 1 Ticker 2
        0  Ticker4  Ticker5
        Output file verification complete. The file meets the
        expected format.
        ',
        'is_success': True,
        'is_finished': True
    }
]
```

## B.2 FAILURE CASE

---

### A failure case of execution error

**Prompt**:

I currently have a structured dataset named date.csv with 6 columns and 5960 rows of data. The first column is named X, the second column is named Y, both of which are structural parameters set for simulation. The following four columns, ff, ft, tt, and tf, represent the simulation results. Now, I need to train a model that can accurately predict the four simulation results for other structural parameters. The model and related content will be saved in the results folder. The model should be saved as "model.pkl" and the final accuracy should be saved in "acc.txt".

**One of the Generated Functions**:

```
# Import necessary libraries
import pandas as pd
import numpy as np
from sklearn.model_selection import train_test_split
from sklearn.preprocessing import StandardScaler
import matplotlib.pyplot as plt
from mpl_toolkits.mplot3d import Axes3D
from scipy.optimize import minimize

# Read in the input data from the 'data.csv' file
data = pd.read_csv('../data.csv')

# Preprocess the data by removing any unnecessary columns and
handling missing values
data.dropna(inplace=True)
X = data.drop('target', axis=1)
y = data['target']
... (omitted)
```

**Execution Error**:

```
-> 6992        raise KeyError(f"{labels[mask].tolist()} not
found in axis")
   6993     indexer = indexer[~mask]
   6994 return self.delete(indexer)

KeyError: "['target'] not found in axis"
```

---

