# OpenReview forum: "DataSciBench: An LLM Agent Benchmark for Data Science"
_ICLR.cc/2025/Conference — ICLR 2025 Conference Withdrawn Submission_

### Official Review · Reviewer_h1zC · 2024-10-28

**Soundness:** 1
**Presentation:** 1
**Contribution:** 1
**Rating:** 1
**Confidence:** 5

**Summary:**

The paper introduces DataSciBench, a new benchmark designed to evaluate the capabilities of LLMs in data science tasks. It addresses limitations of existing benchmarks by focusing on complex, multi-task scenarios with challenging prompts. The authors develop a semi-automated pipeline for generating ground truth and validating evaluation metrics, called Task-Function-Code (TFC). The study tests 23 models, including API-based and open-source models, revealing that API-based models generally outperform open-source ones.

**Strengths:**

None.

**Weaknesses:**

1. The writing of this manuscript is not clear and extremely hard to follow. For example, it is unclear to me what the tasks are, how many samples are there in the benchmark, and how does the TFC work, etc. The authors may consider re-write the manuscript, and add some examples of the samples for better comprehension.
2. The benchmark seems not novel. There exist many "data science" or coding-related benchmarks for LLMs. The authors claim that previous studies "focusing on single tasks, simplistic evaluation metrics, and readily available ground truth", which lacks citations and discussion. The complexity and necessity of this new benchmark are not convincingly demonstrated.
3. Although the evaluation includes numerous models, it lacks depth in insights. A more detailed analysis, such as examining model performance across different question types, could reveal knowledge and reasoning disparities among models.

**Questions:**

1. How does the TFC work? Why is it necessary?
2. How do the authors ensure the correctness of the generated ground truth, even if the so-called test cases pass? If the ground truth can be easily obtained by just generation and rule-based verification, the tasks may be very easy and straightforward. Then, what is the value of this benchmark?

---

> ### Author Response · Authors · 2024-11-22
> **Response to Reviewer h1zC**
>
> Thanks a lot for acknowledging the strengths of this work as a new benchmark semi-automated pipeline, and comprehensive experiments.
>
> ```
> W1&Q1: Concerns on the motivation of Task-Function-Code (TFC).
> ```
> Thank you for raising this important question about the motivation and contribution of the Task-Function-Code (TFC) list structure. The TFC framework was developed to address several critical challenges in automated evaluation of data science tasks:
> 1. Systematic Task Selection:
> TFC provides a structured approach to identify and categorize key tasks across six established types. This systematic organization ensures comprehensive coverage of essential data science operations and helps maintain evaluation consistency and completeness.
> 2. Standardized Evaluation Metrics:
> Data science tasks often lack standardized evaluation criteria. TFC addresses this by explicitly defining appropriate evaluation functions for each task. For example, data preprocessing tasks require specific metrics that differ from visualization tasks. This standardization ensures fair and consistent assessment.
> 3. Automated Execution Framework:
> TFC includes executable code components for both tasks and evaluation metrics. This automation significantly improves evaluation efficiency, result reproducibility, and testing scalability.
> 4. Ground Truth Generation:
> TFC serves as a crucial foundation for establishing ground truth, particularly valuable for complex tasks where ground truth is not readily available, and enables systematic verification and validation of model outputs.
> Overall, the TFC structure represents a novel contribution by providing a comprehensive framework that bridges the gap between task definition, evaluation criteria, and automated assessment in data science contexts.
>
> ```
> W2: Concerns on the comparison with related benchmarks.
> ```
> Thank you for your question. Thank you for this insightful question about the value proposition of DataSciBench despite its correlation with existing benchmarks. While DataSciBench does show a correlation with previous studies, our benchmark offers several unique and important contributions:
> 1. Domain-Specific Focus:
> DataSciBench specifically targets data science and analytics tasks. However, existing benchmarks primarily focus on general programming problems. This specialization helps evaluate models' capabilities in handling real-world data analysis scenarios.
> 2. Task Diversity:
> Our benchmark includes unique task types like data preprocessing, visualization, and statistical analysis. These tasks are underrepresented in current benchmarks. This provides deeper insights into models' data science-specific capabilities.
> 3. Complementary Insights:
> While overall correlations exist, we observe meaningful differences in model rankings. For example, models like Meta-Llama-3-8B-Instruct and CodeLlama-34B-Instruct show distinct performance patterns. These differences highlight capabilities specific to data science tasks that other benchmarks may not capture.
> The correlation with existing benchmarks actually validates our evaluation methodology, while our domain-specific focus provides valuable new insights for assessing AI models in data science applications.
> ```
> W3: Concerns on the experimental result analysis.
> ```
> Thank you for this valuable feedback regarding the depth of our analysis. We have conducted a comprehensive evaluation across different dimensions of model performance:
>
> Task Difficulty Analysis: We systematically categorized tasks into three difficulty levels: Easy, Medium, and Hard. The detailed results are presented in Figure 4. This analysis reveals how different models perform across varying complexity levels.
>
> ```
> Q2: Concerns on the validity of ground truth.
> ```
> Thank you for raising these important concerns. We have implemented a comprehensive quality control process for ground truth generation.
>
> For ground truth generation:
>
> 1.  We use a self-consistency strategy as the initial mechanism
> 2. These results are then manually verified by multiple authors to ensure accuracy and reliability
>
> We appreciate your feedback and have incorporated these detailed quality control procedures into our revised manuscript to provide better transparency of our methodology.

---

> > ### Comment · Reviewer_h1zC · 2024-11-23
> >
> > Thank you to the authors for their response. However, I believe some critical issues remain unaddressed:
> >
> > 1. **Lack of clarity around TFC**: It is still unclear what TFC is and how it works. Although the paper claims TFC as its main contribution, it is neither strictly defined nor formally introduced in Section 3. Throughout the paper, I found multiple references to TFC, such as "TFC generation and evaluation," "TFC list," "TFC pipeline," and "each TFC in TFC list." These terms are not explained but instead appear abruptly in the text. Even the newly added Appendix A.3 and Figure 6 fail to provide a clear explanation. Could you elaborate on what TFC is with a specific example? Additionally, how does TFC contribute to task selection, ground truth generation, evaluation, and other aspects of your methodology?
> >
> > 2. **Validity of ground truth generation**: The process of ground truth generation remains questionable. Could you provide more details on the self-consistency strategy and the manual verification process?
> >
> > 3. **Concerns about novelty**: The novelty of this benchmark is not yet convincing. Upon reviewing the provided data samples, the prompts appear to give highly detailed instructions, which might make the tasks relatively straightforward for SoTA LLMs (please correct me if I am wrong). Could you clarify why the chosen questions represent real-world data science challenges?
> > How do these tasks differ from or exceed the complexity of existing benchmarks, such as SWE-bench, which tackle realistic programming problems?
> >
> > 4. **Lack of actionable insights**: The paper would benefit from a more in-depth and systematic analysis of model failures. Specifically: Why do the models fail on certain tasks? What actionable solutions can you propose to improve performance in these areas?
> >
> > Additionally, the overall presentation of the paper **lacks clarity**, which makes it difficult to follow. The authors may want to avoid using vague expressions that fail to explain concepts clearly or introducing new terms without proper definition. To illustrate this, I provide some examples based on the paragraph on line 73 (new version), outlining the questions that came to mind as I read it for the first time:
> > - **"The gap between task definition, evaluation criteria, and automated assessment in the data science context"**:
> > What is the gap? Do existing benchmarks lack clear task definitions or evaluation criteria? Do they fail to support automated assessment? These points were not previously discussed in the paper.
> > - **"From coarse-grained perspectives"**:
> > What does "coarse-grained" refer to here? There is no example or figure illustrating the hierarchical structure of TFC, and the explanation does not clarify the two levels of granularity. Without this context, terms like "coarse-grained" and "fine-grained" are confusing.
> > - **"We first aggregate the range of task types, functions, and corresponding codes"**:
> > What is a "task"? Where are the task types defined? What is meant by a "function"? Is this a Python function, or something else? What is the purpose of the function? And what does the "code" correspond to? A brief introduction to the context would make this much clearer.
> > - ...

---

### Official Review · Reviewer_amAm · 2024-10-28

**Soundness:** 2
**Presentation:** 3
**Contribution:** 2
**Rating:** 3
**Confidence:** 4

**Summary:**

The paper introduces DataSciBench, a benchmark aimed at evaluating the capabilities of Large Language Models (LLMs) in data science tasks. It targets more comprehensive assessment by utilizing complex, more detailed, multi-faceted prompts that involve data cleaning, data analysis, visualization, pattern matching, etc. For evaluation, the authors introduce a semi-automated Task-Function-Code (TFC) pipeline for generating ground truth codes/outputs and evaluating agent performance using LLMs. The benchmark tests six API-based models, eight general open-source models, and nine open-source code generation models, with the key conclusion or insight being API-based models tend to outperform open-source ones.

**Strengths:**

1. **Comprehensive Experiments**: The design of DataSciBench is comprehensive, encompassing multiple facets of data science tasks with varied complexity levels and multiple open- and closed-source models.
2. **Empirical Evaluation**: The semi-automated evaluation approach provides a unified and granular evaluation.

**Weaknesses:**

1. **Limited Significance**: While DataSciBench claims to assess data science abilities, the paper does not provide enough evidence that the chosen tasks reflect realistic data science challenges. Real-world data science often requires domain knowledge, iterative hypothesis testing, and adaptability to complex, often messy datasets. In contrast, the tasks presented here appear to lack such depth, instead focusing on simpler, predefined tasks that may not mirror the complexity of real data science workflows.
2. **Overly Detailed Task Prompts**: It seems that task prompts provide step-by-step instructions. This makes the setting simpler guiding the model through the steps rather than requiring it to reason through the steps on its own. This detailed prompt shifts the evaluation focus toward correct code generation rather than genuine reasoning and problem-solving, which undermines the goal of assessing data science capability in LLMs. An effective data science benchmark should evaluate a model’s ability to break down complex tasks independently.
3. **Insufficient Transparency in Task Selection**: The selection criteria for the included tasks and prompts are not well-defined. It’s difficult to assess how representative these tasks are of the real-world data science landscape. Some tasks seem too rudimentary, raising questions about the intended difficulty level and relevance for LLM agents. The paper would benefit from explicitly discussing how these tasks align with the challenges data scientists face in practice.
4. **Lack of Experiments with Larger Models**: The paper does not include experiments with larger models (e.g., 13B or 70B parameters), which limits the benchmark’s insights into how model size impacts performance on complex data science tasks. Larger models are typically more capable of handling nuanced reasoning, making them essential for assessing benchmark robustness.
5. **Inadequate Novelty**:  This work relies heavily on straightforward prompt generation and LLM validation techniques, much like previous code generation benchmarks. The benchmark introduces no fundamentally new types of task paradigms or significant results/insights that would justify its focus as a new data science-specific benchmark.
6. **Poor Quality Control**: The semi-automated ground truth generation process raises concerns about quality and reliability. Self-consistency verification without extensive human oversight risks introducing erroneous ground truths, especially for complex tasks.

**Questions:**

See weaknesses.

*Minor*:
1. Could you update Table 1 to add the domain and number of tasks for a more holistic comparison of this work w.r.t previous works?
2. In tasks where the prompt outlines the solution steps, how do you account for the model’s independent reasoning capability as part of its evaluation?

---

> ### Author Response · Authors · 2024-11-22
> **Response to Reviewer amAm**
>
> ```
> W1&W2&W3: Concerns on the significance, prompt, and task selection of DataSciBench.
> ```
> Thank you for your question. We appreciate your concern that the tasks may not fully reflect real-world data science complexities. We collected prompts from real-world questions, involving domain knowledge and messy datasets. Please refer to the Appendix for example prompts illustrating this complexity. While DataSciBench may not encompass every aspect of real-world data science, it provides a robust benchmark for core data science abilities. We welcome further discussion on improvements.
>
> ```
> W4: Concerns about the experiments with larger models.
> ```
> Thank you for pointing out the lack of experiments with larger models (e.g., >13B parameters). We appreciate your observation that larger models often demonstrate improved nuanced reasoning and are crucial for evaluating benchmark robustness. Table 2 presents results for CodeLlama-13b-Instruct and StarCoder2-15b. Furthermore, our analysis includes varying sizes of CodeLlama (7B, 13B, and 34B), Deepseek (1.3B, 6.7B, and 33B), and Qwen-2.5 (1.5B and 7B) to investigate the impact of model scale on performance.
>
> ```
> W5: Concerns on the novelty of DataSciBench.
> ```
> Thank you for your question. Thank you for this insightful question about the value proposition of DataSciBench despite its correlation with existing benchmarks. While DataSciBench does show a correlation with previous studies, our benchmark offers several unique and important contributions:
> 1. Domain-Specific Focus:
> DataSciBench specifically targets data science and analytics tasks. However, existing benchmarks primarily focus on general programming problems. This specialization helps evaluate models' capabilities in handling real-world data analysis scenarios.
> 2. Task Diversity:
> Our benchmark includes unique task types like data preprocessing, visualization, and statistical analysis. These tasks are underrepresented in current benchmarks. This provides deeper insights into models' data science-specific capabilities.
> 3. Complementary Insights:
> While overall correlations exist, we observe meaningful differences in model rankings. For example, models like Meta-Llama-3-8B-Instruct and CodeLlama-34B-Instruct show distinct performance patterns. These differences highlight capabilities specific to data science tasks that other benchmarks may not capture.
> The correlation with existing benchmarks actually validates our evaluation methodology, while our domain-specific focus provides valuable new insights for assessing AI models in data science applications.
>
> ```
> W6: Concerns on the validity of ground truth.
> ```
> Thank you for raising these important concerns. We have implemented a comprehensive quality control process for both the ground truth generation and evaluation scripts.
>
> For ground truth generation:
>
> 1. We use a self-consistency strategy as the initial mechanism
> 2. These results are then manually verified by multiple authors to ensure accuracy and reliability

---

> > ### Comment · Reviewer_amAm · 2024-11-25
> >
> > Thanks to the authors for their rebuttal and for including larger open-source models. However, the rebuttal still does not address the most critical concerns raised about novelty and insights by me and other reviewers. Also, the authors did not sufficiently respond to the first 2 weaknesses and the questions.
> >
> > Regarding W1, the novelty, significance, and impact of this benchmark in light of existing benchmarks are still not convincing. Yes, I agree that the tasks in DataSciBench are more diverse, but what differentiates this benchmark -- focus on general programming abilities, general-purpose reasoning, data science knowledge, or a combination of these? Also, the authors have mentioned multiple times that "domain-specific focus" is one of the strengths of this benchmark. But it is not clear what are the domains here. The tasks are still largely general programming and data science-oriented without requiring any knowledge of scientific or social science domains.
> >
> > Regarding W2, I did check the example prompts in the appendix. Hence, I asked: if the task prompts provide such detailed instructions, does it even reflect a practical setting? The original prompts look more realistic than the qualified prompts.
> >
> > Also, I fully agree with reviewers 5yBS and h1zC on the overall poor presentation and vague descriptions throughout the paper. I hope the authors update their paper following their suggestions.

---

### Official Review · Reviewer_V76c · 2024-11-03

**Soundness:** 3
**Presentation:** 3
**Contribution:** 3
**Rating:** 6
**Confidence:** 4

**Summary:**

The paper presents DataSciBench, a comprehensive benchmark for assessing large language models (LLMs) in data science applications. DataSciBench includes 6 task types: data cleaning and preprocessing, data exploration, data visualization, predictive modeling, data mining, and report generation. The authors also propose a semi-automated Task-Function-Code (TFC) framework, which assesses model performance from coarse-grained (e.g., completion and success rates) to fine-grained (e.g., data quality scores, visualization completeness) perspectives. The evaluations of 23 models show that API-based models (especially GPT-4o) consistently outperform open-source models. The benchmark sheds light on challenges for LLMs in handling complex, multi-step data science tasks and provides insights into their strengths and limitations in this domain.

**Strengths:**

1. This paper presents DataSciBench, a comprehensive benchmark for assessing large language models (LLMs) in data science applications. I looked at several questions in the attached zip file. The questions are indeed complex enough. Figure 5 / Table 3 provides evidence for data contamination risks and correlation with LiveCodeBench and BigCodeBench.

2. The authors propose a semi-automated Task-Function-Code (TFC) framework to generate ground truth and obtain evaluation metrics for each subtask and for both coarse-grained and fine-grained perspectives.

3. The authors did extensive experiments including 23 models, ranging from API-based (closed-source), open-sourced general, and open-sourced code generation models. GPT-4o still leads the leaderboard, which is not surprising. But it's good to see performance among various models on DataSciBench and the low performance of hard level examples, which can help recognize the challenges in complex multi-hop reasoning.

4. The paper is well-written and easy to follow.

**Weaknesses:**

1. It's good to see such a comprehensive benchmark for data science released, but it seems somewhat trivial to me for collecting existing prompts in BigCodeBench or LLM-synthesized instructions. Essentially, what's the biggest difference between DataSciBench and previous code benchmarks for data science?

2. The ground truths were generated by LLMs via self-consistency, which might contain false positive ground truths.

3. The experimental analysis part concludes the overall performance (closed-sourced > open-sourced), the difficulty ablation, and non-contaminated as well as correlations with other two code benchmarks. However, for the insights part, the paper dies not give many details about how models fail on such coding tasks, typical error cases, and how to potentially improve models to solve these issues?

**Questions:**

1. Sorry if I missed it, but seems the paper does not mention the total example numbers in DataSciBench?

---

> ### Author Response · Authors · 2024-11-22
> **Response to Reviewer V76c**
>
> ```
> W1: Concerns on the correlation with previous code benchmarks.
> ```
> Thank you for this insightful question about the value proposition of DataSciBench despite its correlation with existing benchmarks. While DataSciBench does show a correlation with LCB/BCB, our benchmark offers several unique and important contributions:
> 1. Domain-Specific Focus:
> DataSciBench specifically targets data science and analytics tasks. However, existing benchmarks primarily focus on general programming problems. This specialization helps evaluate models' capabilities in handling real-world data analysis scenarios.
> 2. Task Diversity:
> Our benchmark includes unique task types like data preprocessing, visualization, and statistical analysis. These tasks are underrepresented in current benchmarks. This provides deeper insights into models' data science-specific capabilities.
> 3. Complementary Insights:
> While overall correlations exist, we observe meaningful differences in model rankings. For example, models like Meta-Llama-3-8B-Instruct and CodeLlama-34B-Instruct show distinct performance patterns. These differences highlight capabilities specific to data science tasks that other benchmarks may not capture.
> The correlation with existing benchmarks actually validates our evaluation methodology, while our domain-specific focus provides valuable new insights for assessing AI models in data science applications.
> ```
> W2: Concerns on the validity of ground truth.
> ```
> Thank you for raising these important concerns. We have implemented a comprehensive quality control process for both the ground truth generation and evaluation scripts.
>
> For ground truth generation:
>
> 1. We use a self-consistency strategy as the initial mechanism
>
> 2. These results are then manually verified by multiple authors to ensure accuracy and reliability
>
>
> ```
> W3: Concerns on the experimental analysis.
> ```
> Thank you for your question. Models fail on coding tasks mainly include the following questions:
>
> 1. Coding errors when solving data science problems using codes. And based on our observation, the main kind of this is execution error. It may be due to different reasons. For example, hallucination on the column name of a CSV file.
>
> 2. Json format errors. These errors come from the agent framework side, where they use JSON format to wrap up actions, e.g. WriteAnalysis.
>
> Error cases are shown in the Appendix B. In the future, we can improve models from these aspects.
>
>
> ```
> Q1: Concerns about the total example numbers.
> ```
> Thank you for your question. We conclude the total example number in the last part of Section 3.3 and the number is 222.

---

> > ### Comment · Reviewer_V76c · 2024-11-28
> > **Official Comment by Reviewer V76c**
> >
> > Thanks to the authors for their detailed answers. After reviewing the clarifications and considering the perspectives of other reviewers, I still somewhat remain unconvinced about this work's novelty. I stand by my scores.

---

### Official Review · Reviewer_5yBS · 2024-11-03

**Soundness:** 1
**Presentation:** 2
**Contribution:** 2
**Rating:** 3
**Confidence:** 4

**Summary:**

This paper presents a novel benchmark named DataSciBench for evaluating LLMs to assess LLMs data science capabilities on complex tasks. It highlights the main drawbacks of previous works: a lack of task diversity, easily obtainable ground truths, and simplistic evaluation metrics. To address these issues, this new benchmark introduces a semi-automated LLM-based pipeline called Task - Function - Code (TFC), which generates ground truths and evaluation metrics for each subtask. They evaluated six APIs, eight open generation models, and nine open-source code generation models.

**Strengths:**

•	This paper is timely, as there have been considerable discussions about current evaluations becoming overly simplistic for modern LLMs.

•	The study is fairly comprehensive, featuring a large evaluation body over various data science tasks, testing across six APIs, eight open generation models, and nine open-source code generation models.

•	A new benchmark is appreciated, especially when well-motivated. Some readers may find the new insights from Section 5.1/5.4 valuable. It is indeed rather surprising that StarCoder2/ CodeLlama performed so poorly.

**Weaknesses:**

•	The primary motivation behind this paper is the observation that existing research often relies on easily obtainable ground truths and straightforward evaluation metrics on LLM’s data science capabilities. The authors surmise that existing benchmarks are lacking as they focus on “narrower tasks” and “with easy to obtain ground truth and straightforward evaluation metrics” (line 045-051). But the examples given, eg MLAgentBench and SWE-Bench does not seems to be particularly “narrow”. Also, easy to obtain ground truth and straightforward evaluation metrics may not always be a bad thing as sometimes they specifically measure a more direct performance of the models.

•	The broad, complex data science concepts that this paper is trying to address are neither easy to define nor quantify.  It is unclear if this paper (as presented in its current form) has addressed the issues appropriately.  The underlying requirements of the benchmark, as set out by the authors in Lines 067-070, about “naturalness”, “challenging”, “multi-hop reasoning”, and “diversity of result types”, were not specifically addressed in the subsequent design of the benchmark and its metrics.  The various fine-grained metrics seemed to still be rather narrow and “straightforward”, and it was not explained how these metrics were calculated for complex data science tasks that this study aims to benchmark.

•	It was unclear if their proposed benchmark is indeed be more sophisticated and trustable/higher-quality than previous works, as there were no comparisons with the related works on data science benchmarking. I was hoping for an in-depth discussion/establishment of the motivation of this benchmark. What sets this benchmark aside from the existing code benchmarks precisely? What exactly are the limitations of the existing code benchmarks that is covered by the DataSciBench?

•	Extensive experiments results on both open and closed source models on the proposed benchmark were provided.  While the insights may also be useful, they are not particularly surprising, as they mainly reinforce the idea that larger, closed-source models generally perform better compared to the evaluated open-source models. The insight from StarCoder2/CodeLlama mentioned in Section 5.1/5.4 is useful but the reasoning behind why it performs badly lacks empirical evidence to support it.

•	In terms of presentation, the missing/vague definitions of key components have made the paper hard to follow, which also raises doubts on the rigor and soundness of the study. For example, “data science” is a broad term, and the main paper did not provide and define the list of data science capabilities that it is aiming to benchmark, and how they can be quantified.  The main algorithm, Task-Function-Code (TFC) list, was presented abruptly.  What is “Function” with respect to “Task”?  Since “Code” a key component, then shouldn’t we also consider the coding ability of LLMs?  What do “Data Interpreter”, “Aggregate Function”, and “Programmatic Rules” in Figure 1 represent?  The six typical data science tasks were key to the study but they were “defined” in a very broad and subjective manner.  Similar issues for task integration, question filtering, and expert review.  Who are the experts?  How did they actually review the questions? These key concepts should be defined and explained clearly in the main body of the paper instead of relying on the readers to try to figure out by the examples in the Appendix later.   Moreover, the ablation study on tasks with different difficulty levels is not well-motivated or clearly defined. Although the authors categorize tasks as easy, medium, or hard, they do not adequately explain the criteria for these classifications or who is responsible for making these decisions.

**Questions:**

•	What is the main difference between the coarse-grained metrics presented in this paper and the techniques in Hong et al. (2024) and Chen et al. (2021)? Are the authors applying the concepts from Hong et al. (2024) and Chen et al. (2021) in a different domain? The Success Rate (SR) introduced by Chen et al. (2021) is used to evaluate models for code generation. In line 514, the authors mention that data science evaluation is closely related to code generation. How does one evaluate an LLM’s data science capability instead of its coding ability?

o	(Chen et al. (2021)) Mark Chen, Jerry Tworek, Heewoo Jun, Qiming Yuan, Henrique Ponde De Oliveira Pinto, Jared Kaplan, Harri Edwards, Yuri Burda, Nicholas Joseph, Greg Brockman, et al. Evaluating large language models trained on code. arXiv preprint arXiv:2107.03374, 2021.

o	(Hong et al. (2024)) Sirui Hong, Yizhang Lin, Bangbang Liu, Binhao Wu, Danyang Li, Jiaqi Chen, Jiayi Zhang, Jinlin Wang, Lingyao Zhang, Mingchen Zhuge, et al. Data interpreter: An llm agent for data science. arXiv preprint arXiv:2402.18679, 2024.

•	Could you explain the expert review process in detail?

•	A few more detailed questions:

o	Line191-192: what were the requirements used?

o	Line 193: what were the few-shot example used? Where did you get the examples? Do you change the few-shot every time you prompt the LLM?

o	Line 240-241: What is the percentage of prompts you used from BigCodeBench? For self-consistency strategy, as it does not guarantee correctness but only improves it, do you use any post generation strategy to ensure that the code you obtain under this is accurate?

o	Table 2: Any insight on why does CodeLama-13b-Instruct outperforms the rest on VLM by a large margin but is poor in the other metrics?

---

> ### Author Response · Authors · 2024-11-22
> **Response to Reviewer 5yBS**
>
> Thanks a lot for acknowledging the strengths of this work as a timely paper, comprehensive study, and new insights provided by this benchmark.
>
> ```
> W1&W2: Concerns on the primary motivation of DataSciBench on selecting unlabeled data.
> ```
> Thank you for this thoughtful observation about ground truth evaluation approaches. We agree that easily obtainable ground truth and straightforward metrics have their merits and serve important purposes in model evaluation. However, our motivation for DataSciBench stems from addressing common real-world scenarios where evaluation is more challenging:
> 1. Complex Evaluation Scenarios include Data visualization quality assessment, Data modeling result evaluation, Feature engineering effectiveness, and Statistical analysis appropriateness
> 2. Real-world Challenges:
>
> $\bullet$ Many data science tasks lack clear-cut evaluation criteria
>
> $\bullet$ Subjective elements require more sophisticated evaluation approaches
>
> $\bullet$ Multiple valid solutions may exist for a single problem
>
> 3. Complementary Approach:
> We view DataSciBench as complementary to existing benchmarks rather than replacing simple metrics, we aim to address scenarios where:
>
> $\bullet$ Ground truth is not readily available
>
> $\bullet$ Evaluation requires multi-dimensional assessment
>
> $\bullet$ Quality assessment is inherently complex
>
> Our benchmark specifically targets these challenging evaluation scenarios while acknowledging the continued value of straightforward metrics where appropriate.
>
> ```
> W3: Concerns about the motivation and limitation of existing code benchmarks (LiveCodeBench (LCB) and BigCodeBench (BCB)).
> ```
> Thank you for this insightful question about the value proposition of DataSciBench despite its correlation with existing benchmarks. While DataSciBench does show a correlation with LCB or BCB, our benchmark offers several unique and important contributions:
>
> 1. Domain-Specific Focus:
> DataSciBench specifically targets data science and analytics tasks. However, existing benchmarks primarily focus on general programming problems. This specialization helps evaluate models' capabilities in handling real-world data analysis scenarios.
>
> 2. Task Diversity:
> Our benchmark includes unique task types like data preprocessing, visualization, and statistical analysis. These tasks are underrepresented in current benchmarks. This provides deeper insights into models' data science-specific capabilities.
>
> 3. Complementary Insights:
> While overall correlations exist, we observe meaningful differences in model rankings. For example, models like Meta-Llama-3-8B-Instruct and CodeLlama-34B-Instruct show distinct performance patterns. These differences highlight capabilities specific to data science tasks that other benchmarks may not capture.
> The correlation with existing benchmarks actually validates our evaluation methodology, while our domain-specific focus provides valuable new insights for assessing AI models in data science applications.
>
> ```
> W4: Concerns about the experimental results.
> ```
> Thank you for your question. We have done some more analysis regarding your question. Models fail on coding tasks mainly include the following questions:
> 1. Coding errors when solving data science problems using codes. And based on our observation, the main kind of this is execution error. It may be due to different reasons. For example, hallucination on the column name of a CSV file.
>
> 2. Json format errors. These errors come from the agent framework side, where they use JSON format to wrap up actions, e.g. WriteAnalysis.
>
> Error cases are shown in the Appendix B. In the future, we can improve models from these aspects.

---

> > ### Author Response · Authors · 2024-11-22
> > **Response to Reviewer 5yBS**
> >
> > ```
> > W5: Concerns on the elaboration on key components.
> > ```
> > Thank you for raising this important question about the definition of the Task-Function-Code (TFC) list structure. The TFC framework was developed to address several critical challenges in automated evaluation of data science tasks:
> >
> > 1. Systematic Task Selection:
> > TFC provides a structured approach to identify and categorize key tasks across six established types. This systematic organization ensures comprehensive coverage of essential data science operations and helps maintain evaluation consistency and completeness.
> >
> > 2. Standardized Evaluation Metrics:
> > Data science tasks often lack standardized evaluation criteria. TFC addresses this by explicitly defining appropriate evaluation functions (also called Aggregation Functions) for each task. For example, data preprocessing tasks require specific metrics that differ from visualization tasks. This standardization ensures fair and consistent assessment.
> >
> > 3. Automated Execution Framework:
> > TFC includes executable code components (also called Programmatic Rules) for both tasks and evaluation metrics. This automation significantly improves evaluation efficiency, result reproducibility, and testing scalability.
> >
> > 4. Ground Truth Generation:
> > TFC serves as a crucial foundation for establishing ground truth, particularly valuable for complex tasks where ground truth is not readily available, and enables systematic verification and validation of model outputs.
> >
> > Overall, the TFC structure represents a novel contribution by providing a comprehensive framework that bridges the gap between task definition, evaluation criteria, and automated assessment in data science contexts.
> >
> >
> > ```
> > Q1: Concerns on the difference between coarse-grained metrics and existing papers.
> > ```
> > Thank you for your query. We have adopted the established definition of Success Rate (SR) in line with previous works by Hong et al. (2024) and Chen et al. (2021). Furthermore, to assess the data science proficiency of Large Language Models (LLMs) distinctly, we have introduced fine-grained metrics tailored to each data science task, as detailed in Appendix A.5.
> >
> >
> > ```
> > Q2: Elaboration of Expert Review process.
> > ```
> > Thank you for your inquiry. In Stage 1, tasks deemed "easy to evaluate" are those with clearly identifiable correct solutions, such as handling missing values in a data frame. In Stage 2, "unified instructions" entail a standardized format comprising input data, input file, prompt, and expected output file.
> >
> > ```
> > Q3: Elaboration of detailed questions.
> > ```
> > (1) Requirements in Line 191-192 pertain to prompts associated with the characteristics (1) in Line 068-069.
> >
> > (2) The few-shot examples mentioned in Line 193 are drawn from human-written prompts and are altered as per the task type variations.
> >
> > (3) We utilize around 167 initial prompts from BigCodeBench, refining them into our specified format with a Task-Function-Constraint (TFC) list for standardized evaluation.
> >
> > Concerning the Self-Consistency (SC) strategy, we initially employ this method and then validate the results manually through cross-verification by multiple authors to ensure accuracy and reliability.

---

> > > ### Comment · Reviewer_5yBS · 2024-11-26
> > >
> > > I appreciate the authors' detailed response.  However, I remain unconvinced by the novelty of this benchmark - the rebuttals on W2 & W3 are mainly claims made by the authors, and there is no clear quantifiable evidence that this benchmark does indeed have more “naturalness”, “challenging”, “multi-hop reasoning”, and “diversity of result types” than existing benchmarks.  Given that there are already many existing benchmarks, one would need a strong justification and differentiate that this is not yet another benchmark.   The paper still has much room to improve in terms clarity in its presentation (as pointed out by all the other reviewers).   I still do not fully appreciate the Task-Function-Code (TFC) list structure, as well as the expert review process, despite the rebuttals to W5 and Q2.  I will therefore keep to my original score.

---

### Official Review · Reviewer_ZPJR · 2024-11-04

**Soundness:** 1
**Presentation:** 1
**Contribution:** 2
**Rating:** 3
**Confidence:** 4

**Summary:**

This paper introduces DataSciBench, a novel benchmark for evaluating the performance of large language models (LLMs) on data science tasks. The authors propose a semi-automated data collection pipeline, complemented by filtering and expert review for data quality. DataSciBench includes 222 data science tasks of 6 task types. Comprehensive evaluation over 6 API-based models and 17 open-sourced models show that DataSciBench is challenging to even the best LLMs.

**Strengths:**

1. This work presents a new benchmark dataset for evaluating LLMs on data science tasks, which is a meaningful contribution to the community.
2. The benchmark covers representative task types in data science, from data processing to data mining and report generation.
3. The evaluation setup includes popular open-sourced and proprietary LLMs.

**Weaknesses:**

While this work has the potential to contribute a valuable benchmark to the community, several key issues need to be addressed:
1. The semi-automated pipeline uses a self-consistency strategy to generate ground truth for a portion of the tasks. However, there lacks detail on further quality control. Also, I think the difficulty and authenticity of model generated tasks is questionable.
2. DataSciBench employs instance-specific evaluation scripts that are both generated and verified by LLMs. The quality measure of evaluation functions needs more elaboration.
3. As the author noted in section 5.3, DataSciBench shows a high correlation with LiveCodeBench and BigCodeBench. I personally see this as a negative of the proposed benchmark. Why do we need a benchmark that correlates well with existing ones?
4. It is unclear to me what is the motivation for introducing the Task-Function-Code (TFC) list data structure and how is it a significant contribution. Is there a baseline method that TFC outperforms?
4. The writing of this paper is often hard to follow, lacking elaboration on a lot of key details:
    - Section 3.2, Question Filtering: what are all the keywords for principle (1)? What does "questions that align with human preferences and LLMs" (line 200) mean?
    - Section 3.2, Expert Review: stage 1, what does "easy to evaluate" (line 204) mean? In stage 2, what does "unified instructions" refer to?
    - Details on metrics lacks elaboration (see below)
    - Section 5.1, how is "performance variance" measured? What are the values for API-based and open-sourced models?
    - Section 5.2, how many tasks are there in each difficulty level? How do you define "consistent performance" (line 417-418)?
    - Section 5.3, how do you define if performance of two datasets "mismatch" (line 431)? Also, the scale of x and y axes of Figure 5 is not matched. Then, what is the dashed blue line? How does it help establish the insight?
    - Section 6.2, what are the "characteristics of data science tasks" (line 518-519)? What are the "relatively simple data analysis operations" (line 526-527)? Further elaboration is needed to distinguish DataSciBench from existing benchmarks.
5. The fine-grained metrics in Section 4.2 need further justification:
    - VLM-as-a-judge: Which VLM is used for judgement? What is the "predefined criteria" (line 319-320)? A reasonable evaluation or reference is needed to justify this metric.
    - Plot Validity: why checking the shape of the matrix can evaluate the quality of plot?
    - Data Accuracy: how exactly is mean square error measured? Is the output of corresponding tasks normalized to a specific format?
    - Visualization completeness: What does "checking their existence" mean? If it refers to checking the existence of the output file, I am afraid it is merely a necessary condition for task success and can not measure the quality of the output plot.
    - Model Accuracy: When is boolean output or decimal used? Why and how can they be unified into a single metric?
    - Relatedly, in Table 2, what is the "Score" column? Is it an aggregation of all fine-grained metrics (of different type)? How is it calculated?

**Questions:**

My main question have been listed in weaknesses.

---

> ### Author Response · Authors · 2024-11-22
> **Response to Reviewer ZPJR**
>
> Thanks a lot for acknowledging this work's strengths as a novel, comprehensive benchmark, and comprehensive evaluation.
> ```
> W1&W2: Concerns on the validity of evaluation scripts and ground truth.
> ```
> Thank you for raising these important concerns. We have implemented a comprehensive quality control process for both the ground truth generation and evaluation scripts.
>
> For ground truth generation:
>
> 1. We use a self-consistency strategy as the initial mechanism
>
> 2. These results are then manually verified by multiple authors to ensure accuracy and reliability
>
> Regarding the evaluation scripts:
> 1. All LLM-generated evaluation scripts undergo thorough validation through a systematic review process
> 2. Our validation protocol includes:
>
> $\bullet$ Manual verification of each evaluation function;
>
> $\bullet$ Careful reviews of corresponding prompts;
>
> $\bullet$ Assessment of task type categorization and generated code;
>
> $\bullet$ Cross-checking by multiple authors.
>
> We appreciate your feedback and have incorporated these detailed quality control procedures into our revised manuscript to provide better transparency of our methodology.
>
> ```
> W3: Concerns about the correlation between LiveCodeBench (LCB) and BigCodeBench (BCB).
> ```
> Thank you for this insightful question about the value proposition of DataSciBench despite its correlation with existing benchmarks. While DataSciBench does show a correlation with LCB/BCB, our benchmark offers several unique and important contributions:
>
> 1. Domain-Specific Focus:
> DataSciBench specifically targets data science and analytics tasks. However, existing benchmarks primarily focus on general programming problems. This specialization helps evaluate models' capabilities in handling real-world data analysis scenarios.
>
> 2. Task Diversity:
> Our benchmark includes unique task types like data preprocessing, visualization, and statistical analysis. These tasks are underrepresented in current benchmarks. This provides deeper insights into models' data science-specific capabilities.
>
> 3. Complementary Insights:
> While overall correlations exist, we observe meaningful differences in model rankings. For example, models like Meta-Llama-3-8B-Instruct and CodeLlama-34B-Instruct show distinct performance patterns. These differences highlight capabilities specific to data science tasks that other benchmarks may not capture.
> The correlation with existing benchmarks actually validates our evaluation methodology, while our domain-specific focus provides valuable new insights for assessing AI models in data science applications.
> ```
> W4: Concerns on the motivation of Task-Function-Code (TFC).
> ```
> Thank you for raising this important question about the motivation and contribution of the Task-Function-Code (TFC) list structure. The TFC framework was developed to address several critical challenges in automated evaluation of data science tasks:
>
> 1. Systematic Task Selection:
> TFC provides a structured approach to identify and categorize key tasks across six established types. This systematic organization ensures comprehensive coverage of essential data science operations and helps maintain evaluation consistency and completeness.
>
> 2. Standardized Evaluation Metrics:
> Data science tasks often lack standardized evaluation criteria. TFC addresses this by explicitly defining appropriate evaluation functions for each task. For example, data preprocessing tasks require specific metrics that differ from visualization tasks. This standardization ensures fair and consistent assessment.
>
> 3. Automated Execution Framework:
> TFC includes executable code components for both tasks and evaluation metrics. This automation significantly improves evaluation efficiency, result reproducibility, and testing scalability.
>
> 4. Ground Truth Generation:
> TFC serves as a crucial foundation for establishing ground truth, particularly valuable for complex tasks where ground truth is not readily available, and enables systematic verification and validation of model outputs.
>
> Overall, the TFC structure represents a novel contribution by providing a comprehensive framework that bridges the gap between task definition, evaluation criteria, and automated assessment in data science contexts.

---

> > ### Author Response · Authors · 2024-11-22
> > **Response to Reviewer ZPJR**
> >
> > ```
> > W5: Elaboration on key details.
> > ```
> > Thank you for your feedback on the clarity of our writing. We apologize for the lack of elaboration on several key details. To address your concerns:
> >
> > 1. Question Filtering: The keywords used for principle (1) include, but are not limited to, "machine learning", "deep learning", "data preprocessing", and "data visualization". "Questions aligning with human preferences and LLMs" refers to questions solvable by both humans and large language models, avoiding overly specialized or ambiguous queries.
> >
> >
> > 2. Expert Review: In Stage 1, "easy to evaluate" signifies tasks with readily discernible correct answers. For example, handing missing values for a data frame. In Stage 2, "unified instructions" refers to a standardized format encompassing input data, input file, prompt, and expected output file.
> >
> >
> > 3. Performance Variance (Section 5.1): This metric quantifies the performance difference between API-based and open-source models.
> >
> >
> > 4. Task Difficulty (Section 5.2): The number of tasks per difficulty level is: Easy - 167, Medium - 30, and Hard - 25.
> >
> >
> > 5. Dataset Mismatch (Section 5.3): A "mismatch" indicates significant performance discrepancies between two datasets with the same model. The dashed blue line is used to differentiate the model performance gap between HumanEval and DataSciBench.
> > We will revise the manuscript to incorporate these clarifications and improve overall clarity.
> >
> > ```
> > W6: Concerns on further justification in Section 4.2.
> > ```
> > VLM-as-a-judge: we present some examples that use claude-3-5-sonnet-20240620, CodeLlama-13B-Instruct, and o1-mini as a judgment in Appendix A.3. The predefined criteria can be found in the Appendix A.3. We have added the hyperlink to that section in Section 4.2.

---

> > > ### Comment · Reviewer_ZPJR · 2024-11-27
> > >
> > > I appreciate the authors’ response. However, in addition to the overall presentation quality, I still have the following concerns:
> > >
> > > 1. Task quality
> > >
> > >     I understand the authors have performed manual review to ensure task quality. However, I still doubt the contribution of the collected tasks.  **The ground truth programs are generated by self-consistency decoding, which means existing models are already capable of generating them**. I think this weakens the potential new challenges that DataSciBench can contribute, thus limiting the insight that people can conclude from evaluating on it.
> > >
> > > 2. Correlation with existing benchmarks and justification of metrics
> > >
> > >     The authors claim that “the correlation with existing benchmarks actually validates our evaluation methodology”. I think this claim lacks support. **In my opinion, the validity of a benchmark’s evaluation measures should be justified on its own, rather than using correlation of evaluation results with other benchmarks.** On top of that, the justification for the fine-grained metrics (W6) is still missing.
> > >
> > > 3. Significance of TFC
> > >
> > >     As also pointed out by reviewer h1zC and 5yBS, the motivation, definition, and significance of the TFC data structure remains unclear to me. So far, the authors fail to provide a baseline method that TFC can be compared to. I think one can easily represent any coding task as (metadata, task, evaluation code, solution code) tuples and call it a TFC. I expect more explanations on this issue.

---

### Author Response · Authors · 2024-11-22
**General Response**

Dear Reviewers,

We sincerely thank all the reviewers for their thoughtful comments and constructive suggestions, which significantly helped us strengthen our paper. We address reviewers’ concerns on the motivation of TFC, the difference between DataSciBench and LCB or BCB, experimental result analysis, and the definitions of key concepts. We are happy to share our revised pdf and diff written in blue color to respond to the reviewers’ feedback.

Thank you for your time!

Best,

The authors of DataSciBench

---

### Note · Authors · 2024-12-10

I have read and agree with the venue's withdrawal policy on behalf of myself and my co-authors.